# Identification of the Primary Factors Determining the Specificity of Human VKORC1 Recognition by Thioredoxin-Fold Proteins

**DOI:** 10.3390/ijms22020802

**Published:** 2021-01-14

**Authors:** Maxim Stolyarchuk, Julie Ledoux, Elodie Maignant, Alain Trouvé, Luba Tchertanov

**Affiliations:** Centre Borelli, CNRS, ENS Paris-Saclay, Université Paris-Saclay, 4 av. des Sciences, F-91190 Gif-sur-Yvette, France; maksim.stoliarchuk@ens-paris-saclay.fr (M.S.); julie.ledoux@ens-paris-saclay.fr (J.L.); elodie.maignant@ens-paris-saclay.fr (E.M.); alain.trouve@ens-paris-saclay.fr (A.T.)

**Keywords:** hVKORC1, Trx-fold proteins, protein folding, dynamics, molecular recognition, thiol–disulphide exchange, protein–protein interactions, PDI–hVKORC1 complex, 3D modelling, molecular dynamics simulation

## Abstract

Redox (reduction–oxidation) reactions control many important biological processes in all organisms, both prokaryotes and eukaryotes. This reaction is usually accomplished by canonical disulphide-based pathways involving a donor enzyme that reduces the oxidised cysteine residues of a target protein, resulting in the cleavage of its disulphide bonds. Focusing on human vitamin K epoxide reductase (hVKORC1) as a target and on four redoxins (protein disulphide isomerase (PDI), endoplasmic reticulum oxidoreductase (ERp18), thioredoxin-related transmembrane protein 1 (Tmx1) and thioredoxin-related transmembrane protein 4 (Tmx4)) as the most probable reducers of VKORC1, a comparative in-silico analysis that concentrates on the similarity and divergence of redoxins in their sequence, secondary and tertiary structure, dynamics, intraprotein interactions and composition of the surface exposed to the target is provided. Similarly, hVKORC1 is analysed in its native state, where two pairs of cysteine residues are covalently linked, forming two disulphide bridges, as a target for Trx-fold proteins. Such analysis is used to derive the putative recognition/binding sites on each isolated protein, and PDI is suggested as the most probable hVKORC1 partner. By probing the alternative orientation of PDI with respect to hVKORC1, the functionally related noncovalent complex formed by hVKORC1 and PDI was found, which is proposed to be a first precursor to probe thiol–disulphide exchange reactions between PDI and hVKORC1.

## 1. Introduction

Thioredoxins (Trxs) are disulphide reductases that are responsible for maintaining proteins in their reduced state inside cells. Trxs are involved in a wide variety of fundamental biological functions ([1] and references herein) and, therefore, are vital for all living cells, from archaebacteria to mammals. The wide variety of Trx reactions is based on their broad substrate specificity and potent capacity to reduce multiple cellular proteins [2]. This broad specificity for thioredoxin and related proteins has made it difficult to distinguish the true physiological partners for the protein from in vitro artefacts.

All membrane-associated Trx proteins possess an active site made up of two vicinal cysteine (C) residues embedded in a conserved CX_1_X_2_C motif. These two cysteines, separated by two residues, play a key role in the transfer of two hydrogen atoms to the oxidised target and the breaking of the Trx–disulphide bond (Figure 1A). This disulphide-relay pathway is accompanied by an electron transfer in the opposite direction. An intermediate state during the electron transfer is a mixed disulphide bond formed by a pair of cysteine residues from two proteins, which can be resolved by the nucleophilic attack of a thiol group from one of the flanking cysteine residues. Through this mechanism, the disulphide is exchanged within one thiol oxidoreductase or between a disulphide donor and a target protein [3]. Thiol–disulphide exchange reactions occur between redox-sensitive biomolecules if donors and acceptors can interact in the appropriate orientations when attacking and leaving groups [4].

The thioredoxin fold is the most common structure found in thiol oxidoreductases; it has been carefully described in [7]. It is illustrated with the crystallographic structure of human protein disulphide isomerase (PDI; Figure 1B), which is the best-characterised enzyme that assists in the process of oxidative folding [8,9]. The PDI structure consists of a central five-stranded propeller with four flanking *α*-helices, an architecture that contains extra regions compared to the classical thioredoxin fold (a four-stranded β-sheet with three *α*-helices formed by about 80 residues).

The dithiol/disulphide group in the CX_1_X_2_C motif, which is located at the head of the αH2 helix, protrudes from the protein surface and is exposed to a solvent. Such a spatial arrangement of the CX_1_X_2_C motif is probably to ensure the full accessibility of the first cysteine, which is required to react with the cysteine residue of a target to accomplish redox processes. It has been reported that the reactive thiolate of this first cysteine can be stabilised by the positive dipole at the head of the αH2 helix and by a network of hydrogen (*H*) bonds that are formed between the thiolate and neighbouring residues presented by the helix-turn structure [10].

In the present study, the focus is on the Trx’s function as a physiological reductant (*H*-donor) of vitamin K epoxide reductase complex 1 (VKORC1). VKORC1 is an endoplasmic reticulum-resident transmembrane protein that is responsible for the activation of vitamin K-dependent proteins, and it is involved in several vital physiological and homeostasis processes [11]. VKOR is the target of oral anticoagulants like warfarin, which dampens coagulation by limiting the supply of vitamin K. Its functional role is a catalyst in the reduction of vitamin K, requiring cooperation with a redox partner that delivers reducing equivalents. A particularly interesting problem is the enzymatic activation of hVKORC1 by the thiol–disulphide exchange. This process involves “molecular recognition” at the highest level required for proton-transfer reactions.

Recently, 3D models of human VKORC1 (hVKORC1) have been reported along with functionally related enzymatic states [6]. The models that were generated for the metastable states of hVKORC1 and their validation through in silico and in vitro screening have led to a conceptually plausible mechanism for enzymatic reactions based on a sequence array of hVKORC1-activated states involved in vitamin K transformation. These results suggest several additional questions, the most important being the real enzymatic machinery of hVKORC1 and its activation. Which Trx-fold protein is a specific proton donor of hVKORC1? What are the factors controlling the specificity of hVKORC1 recognition by the Trx protein? What is the exact role of thioredoxin(s) in initiating hVKORC1 reduction?

Since the physiological reductant of hVKORC1 has not yet been identified, initial exploration was made of four human redoxin proteins, namely, protein disulphide isomerase (PDI), endoplasmic reticulum oxidoreductase (ERp18), thioredoxin-related transmembrane protein 1 (Tmx1) and thioredoxin-related transmembrane protein 4 (Tmx4), reported as the most probable *H*-donors of VKOR [12,13]. These proteins have distinct compositions for the active site CX_1_X_2_C-CGHC in PDI, CGAC in ERp18, CPAC in Tmx1 and CPSC in Tmx4—and they show broad but distinct substrate specificity. The nature of this specificity is the main focus of this work. In order to evaluate the one most likely to reduce hVKORC1, a detailed comparison of these redoxins was first provided at different levels of the protein’s organisation—sequence, secondary and tertiary structure, intrinsical dynamics, intraprotein interactions governing structural and conformation properties, and composition of the surface exposed to the targets. Second, hVKORC1 in its native state, in which two pairs of cysteine residues form two disulphide bridges (Figure 1D), was studied as a target of Trx-fold proteins in order to identify the anchor site(s) that enable it to recognise/bind its Trx effector. Finally, modelling of the complex formed by hVKORC1 and PDI, which was suggested as the most probable partner of VKORC1, was carried out using the PDI fragments predicted to be “interacting” as a guide and the VKOR structure from Synechococcus sp (bVKOR; PDB ID: 4nv5; Figure 1E; see [14]) as an initial reference. The model of the molecular noncovalent complex formed by PDI and hVKORC1 (PDI–hVKORC1) is proposed as the first useful human precursor for the probing of thiol–disulphide exchange reactions between redoxins as an *H*-donor and hVKORC1 as a substrate.

This study principally leans on molecular dynamics (MD) simulation of the chosen Trx-fold proteins in the reduced state, of human VKORC1 in the inactive (oxidised) state, and on the modelling of the molecular noncovalently bound complex formed by hVKORC1 and PDI. It is suggested that a careful analysis of the simulation data will deliver quantitative and qualitative metrics to shed light on the following questions: (i) Are the 1D, 2D and 3D properties and the dynamic features good indicators for the prediction of the protein fragments participating in hVKORC1 recognition by a Trx? (ii) From the in-silico study of proteins, is it possible to predict which of them is the most likely partner of VKORC1? (iii) How do the predicted results correspond to a model of the complex formed by VKORC1 and its possible partner?

A central goal of this study is to understand, at the atomistic level, the recognition mechanisms between Trx and hVKORC1 (a process preceding the electrons’ transfer reaction) and, thereby, identify shared vulnerable sites that can be targeted with anti-hVKORC1 or anti-Trx therapeutics.

## 2. Results

### 2.1. The Trx-Fold Proteins as Possible Partners of VKORC1

#### 2.1.1. Sequences and Structural Data

Structures of PDI (PDB ID: 4ekz; [9]), ERp18 (PDB ID: 1sen; [15]) and Tmx1 (PDB ID: 1x5e; [5]) were used to extract the coordinates of a domain containing the CX_1_X_2_C motif (Appendix A, Appendix A). This domain was chosen for the study of all proteins because ERp18, Tmx1 and Tmx4 proteins are only constituted of one Trx-fold domain **a**. The sequences of the four selected Trx proteins show a low identity/similarity (Figure 1C, Appendix A) along with the best scores for Tmx1 and Tmx4 (47/68%). The ERp18 sequence differs most from those of PDI, Tmx1 and Tmx4 (23/38%, 15/23% and 15/23%, respectively). A 3D model of Tmx4 was built from Q9H1E5 (https://www.uniprot.org/uniprot/), with the Tmx1 structure as a template.

The ERp18, PDI and Tmx1 empirical structures and the Tmx4 homology model were optimised (when necessary) to obtain a CX_1_X_2_C motif in the reduced state. These were then used for the conventional MD simulations (two 500-ns trajectories for each protein), running under strictly identical conditions.

#### 2.1.2. General Characterisation of Trx-Fold Proteins Using MD Simulations

The global stability of each Trx-fold protein over the course of a simulation was estimated using the root mean square deviation (RMSD) that showed (i) similar behaviour for the same protein among both MD replicas and (ii) significant disparity between the different proteins (Figure 2A). Comparable RMSDs for PDI over each replica and between replicas characterise a highly stable protein structure during the simulation. Similar to PDI, the RMSD values for ERp18, Tmx1 and Tmx4 varied within a narrow range after elimination of the largest amplitude *N*/*C*-terminal residues. This demonstrates the good structural stability of each Trx, which is a quality that is typical of well-organised folded regular proteins.

Indeed, in all studied Trx-fold proteins, the properly ordered secondary structures (SS or 2D structure) were shown to be long-lived α-helices and β-strands. These ordered structures are interconnected by coiled linkers to form a stable globular 3D arrangement that is described as a four- or five-stranded antiparallel β-sheet sandwiched between four α-helix-bundle structures, which is an archetypical fold of the Trx family of proteins (Figure 2B). Similar to the RMSDs, the root mean square fluctuations (RMSFs) agree well between the pair of replicas for protein (Figure 2C and Appendix A). The most pronounced difference in RMSFs between the two replicas is only observed in ERp18, in which β5 is partially unfolded and the L7 and L8 loops are joined together, resulting in large fluctuations.

Further characterisation of each protein and a comparison between the proteins is frequently completed by the observations obtained for a randomly chosen single trajectory or concatenated data. This is because the RMSDs and RMSFs in both replicas of each protein display comparable profiles and a similar range of values, and the 2D and 3D structures of each protein are perfectly matched (the RMSD values between the average structures of replicas 1 and 2 are less than 0.5 Å; Figure 2). The exception is PDI, in which the αH2-helix showed a different length over two replicas that was caused by the distant fold of its *N*-terminal.

How different are the 2D and 3D structures for the four proteins? The organised secondary structures, α- and 3_10_-helices and β-strands, involve 55%, 60%, 60% and 56% of the residues in ERp18, PDI, Tmx1 and Tmx4, respectively, where the helical and β-strand fold portions vary from 36% to 42% and from 13% to 22% of total folding, respectively. Although all ordered 2D structures (helices and strands) are generally conserved across the studied proteins, their positions, lengths and qualities (e.g., α- or 3_10_-helix) are slightly different (Figure 2 and Appendix A).

The helical fold of each protein is represented by α-helices of different length (of 7–18 residues) and by 3_10_-helices that consist of 3–4 residues. H1, which is a long-lived α-helix in ERp18 and PDI, is transient and converts between α- and 3_10_-helices in Tmx1 and Tmx4. H2, which is the longest α-helix (14–18 residues) that contains the CX_1_X_2_C motif at its *N*-extremity, is generally conserved in all proteins; however, it may be partially split into two helices (ERp18) or reduced in size (PDI). The folding of the CX_1_X_2_C motif is different in the four proteins, and this represents a part of the regular α-helix (ERp18 and Tmx1), a transient helix fluctuating between α- and 3_10_-helices or/and a turn (PDI) and a coiled structure (Tmx4). In ERp18, H3 consists of a pair of short 3_10_-helices, while in the other proteins, it is a single and stable α-helix. H4 is a long and stable α-helix in ERP18 and PDI, while in Tmx1 and Tmx4, it is folded as a shorter α-helix and is joined to a 3_10_-helix.

This analysis illustrates that although the studied proteins share a similar structure, their folding is noticeably different; this reflects their sequence-dependent character.

Additionally, the atomistic RMS fluctuations of the studied proteins show (i) minimal RMSF values for all β-strands forming the antiparallel β-sheet in all proteins, while the helices may have discernable fluctuations (e.g., αH2 and αH3 in Tmx1, and αH2 in Tmx4), and, as was expected, (ii) strong differences in the fluctuations of the coiled linkers (Figure 2C). These linkers, which interconnect the core β-stands and the surrounding α-helices, are the most variable elements in the studied proteins in terms of sequence composition, length and conformation. It is also noted that moderate (in the order of 1.5–2.5 Å) but systematically observed fluctuations of fragment L5-αH3-L6 arose in all studied proteins. This fragment is structurally adjacent to the CX_1_X_2_C motif and may play a role in thiol–disulphide exchange reactions.

#### 2.1.3. Intrinsic Motion and Its Interdependence on Trx-Folded Proteins

Since a protein’s dynamics influence its functional properties, intrinsic motions of Trx-fold proteins were compared. First, a cross-correlation map was computed for all Cα-atom pairs of each protein (Figure 3A). The positively correlated motion of β2-, β3- and β4-strands, which was observed in each studied protein, reflects their concerted movement in the β-barrel. To equilibrate structural stability, the other fragments in Trx-fold proteins display a motion that tends to correlate negatively. As such, in ERp18, in addition to the β-barrel coupled motion, the structural moieties with the strongest correlation are L7 and αH4. In PDI, a regular fractal-like pattern shows the correlated motion of αH1 with αH2 and L7 and αH3 with L7 and L8. In Tmx1, the coupled motion is observed between the αH2-helix and the αH4-helix and between the β2-stand and the αH3-helix. Tmx4 demonstrates correlated motions between the αH2-helix and the β3-strand and between the αH3-helix and β4/β5-strands.

The collective motion of Trx-fold proteins and its impact on their conformational properties was studied using a principal component analysis (PCA). The principal components (PCs) were determined, and the MD conformations for each protein were projected onto the PC subspace formed by the first two and first three eigenvectors. This indicated that Tmx1 (green) and Tmx4 (blue) conformations were grouped in a unique compact region for each protein, and these regions were perfectly superimposed for both proteins, while the conformations of PDI (red) and ERp18 (yellow) were trapped in two or three separate regions that were located in a slightly enlarged space (Figure 3B). Randomly selected conformations from the distinct regions in the projection of the first two PCA modes showed that their conformational difference is mainly associated with a motion that leads to a slight skew of the H5-helix and displacement of the H3-helix in ERp18 and a disparity in the H2-helix length in PDI.

From the ten calculated PCA modes describing ~95% of total backbone fluctuations of each Trx-fold protein, the first two most dominant modes were used to illustrate ample collective movements qualitatively (Figure 3C). The PCA modes of the Trx-fold proteins reveal the essential mobility of their fragments, which is either similar in the four proteins or has different features for a given protein. For instance, in ERp18, the greatest mobility is observed for L7 and L8 loops that are joined together due to the unfolding of the β5-strand. In PDI and Tmx4, the L7 and L8 loops are well separated by the β5-strand, but each of them shows the coupled motion of a large amplitude. Uniquely, in Tmx1, the αH3-helix and its joint L5 loop display a high amplitude motion. In PDI, Trx1 and Trx4, the collective motion of the H2-helix and joint L3 loop is comparable in amplitude but differs in directions.

#### 2.1.4. Focus on the Region of Trx-Fold Proteins Potentially Involved in Target Recognition and/or Electron Transfer Reaction

To compare the four Trx-like proteins regarded as probable functional effectors of hVKORC1, the focus was on two fragments that may be involved in target recognition and/or electron transfer reaction. The first fragment, **F1**, comprises L3 and an *N*-extremity of αH2-helix that includes the CX_1_X_2_C motif and the second, **F2**, which is structurally adjacent to the CX_1_X_2_C motif, is composed of L5-αH3-L6. Both fragments form a frontal region that is exposed to the solvent in each Trx-fold protein, which may interact directly with a target during the electron-exchange process, similar to a bacterial protein containing a Trx-fold domain that is covalently bonded to VKOR (Figure 1E). The delimiting of these two regions is very approximate because the sequences and 2D structures of the studied proteins show significant differences. To have segments of a comparable length in different proteins, the boundaries of fragments were chosen so that their lengths were equal (17 residues; Figure 4).

The **F1** region in PDI, Tmx1 and Tmx4 is similarly initiated by tyrosine, which is the residue reported to be a breaker of secondary structures, while in ERp18, the role of “a breaker” is given to histidine, followed by lysine, which are amino acids that are more likely to be present in disordered regions [16,17]. The following residues of the L3-loop, a pair of hydrophobic residues (APs), are perfectly conserved in PDI, Tmx1 and Tmx4, while in ERp18, these positions are occupied by positively charged and polar residues (KSs). Furthermore, the specific CX_1_X_2_C motif for each studied protein is preceded by tryptophan (W), which is a highly conserved residue in the four proteins. Tryptophan is an amphipathic residue that, similar to tyrosine, is often found at the surface of proteins and is sometimes classified as polar.

It is suggested that the **F1** region of Trx-folded proteins, which contains the CX_1_X_2_C motif, contributes to redox reactions rather than target recognition. Nevertheless, a double action of the **F1** fragment as both redox agent and recognition platform for a target has not been excluded.

The second surface region of Trx-fold proteins, **F2**, which is in the proximity of the CX_1_X_2_C motif, consists of the αH3-helix and its two adjacent loops, L5 and L6. This fragment shows a negligible or no similarity/identity between the four proteins and, thus, may convey the highest degree of specificity in the discrimination/recognition of a partner. The most critical difference consists of the sequence composition of the L5 loop and the αH3-helix and the length of the αH3-helix. In ERp18, a set of five negatively charged amino acids (EDEEEs), which are positioned on the L5 loop and the αH3-helix, are separated by proline (P) and lysine (K) from the other three negatively charged amino acids (DEDs). This promotes a breakup of the H3-helix into two smaller 3_10_-helices. In the other proteins, the number of negatively charged residues in this region is diminished to four in PDI and one in Tmx1 and Tmx4. The two last proteins, Tmx1 and Tmx4, have the same αH3-helix content and differ only in the combination of amino acids in L5. Despite a great difference in the αH3-helix composition of PDI compared to that of Tmx1 and Tmx2, the length of the helix in the three proteins is equivalent (6 aas). In all studied proteins, the short loop L5 contains at least one negatively charged residue and one polar residue, while the extended L6 loop is mainly composed of hydrophobic residues enriched by one or two polar residues with an inserted charged amino acid (the negative in ERp18 and the positive in PDI).

As the αH3-helix is moving considerably in Tmx1 and moderately in the other proteins (Figure 3C), we suggest that the αH3-helix can adapt its orientation to get the best position with respect to the target and, together with its joint loops, L5 and L6, is able to build the recognition (docking) site(s) for target accommodation. The **F2** region is the most dissimilar fragment in the studied proteins, and it has a sequence composed of hydrophobic stretches folded into a polar lipid environment. **F2** also contains polar and charged residues required for stretches of sequence that are exposed to a solvent in cytosolic or extracellular environments [18]. Therefore, **F2**, which is positioned in the proximity of the CX_1_X_2_C motif, is a fragment of a Trx-fold protein that can contribute to VKORC1 recognition.

#### 2.1.5. Geometry of the CX_1_X_2_C Motif

Focusing on the CX_1_X_2_C motif, a key agent in thiol–disulphide exchange reactions, its geometry was characterised in each Trx-fold protein. It was observed that structurally, the CX_1_X_2_C motif constitutes either a part of the αH2-helix (in ERp18 and Trx1), which is transient in PDI, or an extension of the L3 loop (in Tmx4). Both cysteine residues that are located on a coil are largely exposed to the solvent, whereas only one cysteine is exposed in the folded CX_1_X_2_C, while the other cysteine is buried in the protein chain.

Surprisingly, the folding of the CX_1_X_2_C (CGAC) motif in the calculated conformations (MD simulation) of ERp18 is coherent with those observed in the experimentally determined structure (PDB ID: 1sen), despite the different protein states, namely, reduced (MD simulation), with two protonated thiol groups, and oxidised (X-ray analysis), in which two deprotonated thiol groups form a disulphide bridge. In both protein states studied by the two different methods, the first cysteine from the CGAC motif is the N-cap residue (the last nonhelical residue) of the α-H2 helix.

The second unexpected observation is connected to the different folding of the CX_1_X_2_C (CGHC) motif in the calculated (MD simulation) and empirical structures (X-ray analysis) of PDI when studied in the same state (reduced). Indeed, the CX_1_X_2_C motif in the crystallographic structure of PDI was reported as folded, with the C37 positioned at the cap of the α-H2 helix (PDB ID: 4ekz), while in the MD conformations, the structure of this motif is transient and alternated between the helical fold (α- or 3_10_-helices) and the turn/coiled structure, demonstrating high conformational plasticity.

The folding of the CX_1_X_2_C motif in Tmx1 (CPAC) in the MD conformations and the NMR structures (PDP ID: 1x5e; both are in a reduced state) is equivalent, with the first cysteine as an *N*-cap residue of the downstream α-H2 helix, similar to ERp18. In Tmx4, a protein with the most similar sequence to Tmx1, the CX_1_X_2_C motif (CPSC) demonstrates a coiled structure. In these two proteins, the conserved proline constitutes the characteristic CPX_2_C motif, and the observed structural differences may be connected either to the X_2_ residue or to the long-distance structural effects.

The geometry of the CX_1_X_2_C motif was described by two metrics: a distance S⋯S′ between the protonated sulphur atoms and a dihedral angle S−Cα−Cα′−S′ (Figure 5A and Appendix A). In proteins ERp18 and Tmx1, the mean value (mv) of these parameters (4 Å and 60°, respectively) describe a synclinal configuration (Prelog–Klyne nomenclature) of the sulphur atoms that is well-conserved over the MD simulations. Nevertheless, a rare but not-negligible number of Tmx1 conformations revealed a syn-periplanar or anticlinal orientation of sulphur atoms that promoted a slight increase in the S⋯S distance. Such restrained geometry of the CX_1_X_2_C motif in Erp18 and Tmx1 is apparently related to its location on the well-folded αH2-helix. By contrast, the CX_1_X_2_C motif located on a coiled L3 loop in Tmx4 stimulates a highly divergent orientation of sulphur atoms, running from syn-periplanar configuration to an antiperiplanar configuration, as was evidenced by a large variation in the dihedral angle S−Cα−Cα−S. The measured metrics, distance S⋯S and dihedral angle in PDI had values close to those in Erp18 and Tmx1. Nevertheless, a large number of conformations displayed a strongly variant geometry, which is similar to Tmx4. Such richness in PDI conformations corresponds to the transient structure of the *N*-terminal of the H2-helix, conversed between the helical fold (α- and 3_10_-helices) and turn structure.

To better characterise the dynamical behaviour of the CX_1_X_2_C motif over two trajectories for each protein and to compare the different proteins, 3D skeletal shape trajectories of the motif’s atoms were described in Kendall’s shape space [19]. For a given integer k, Kendall’s shape space is the manifold of dimension 3k − 7 dimension of all possible configurations of *k* atoms in R^3^ considered up to a rigid transformation (translation, rotation and scaling). It has a Riemannian structure with a computable geodesic distance. The framework allows the use of geometric statistics and dimension reduction methods like multidimensional scaling (MDS) to analyse the shape trajectories [20]. These methods offer various ways of visualising all the data in a common space, summarizing them with a reduced number of variables and comparing them to each other. A tetrahedron, defined for the S- and Cα-atoms of two cysteine residues, C37 and C40, was extracted from conformations over MD simulations (Figure 5C). The four proteins can be condensed in two major groups that are weakly overlapping (clearly visible in the 3D view): ERp18 and TMX1 on the one hand, PDI and TMX4 on the other hand, the latter group displaying a larger shape variation.

This analysis is illustrated by the superposition of the thiol groups (Cα-S-H) from the CX_1_X_2_C motif of the MD conformations for each protein (Figure 5B). The orientation of the thiol groups favours H-bond interaction (S−H⋯S) only in ERp18 and some Tmx1 conformations. In PDI, both the thiol groups are shown to have the most variant orientation within a group and between groups, which reflects their high mobility.

The *H*-bond between the sulphur atoms of each cysteine is characterised for two cases: (**1**) the S-atom from C37 is the H-donor to the S-atom of C40, and (**2**) the S-atom from C40 is the H-donor to the S-atom of C37 (Figure 5D and Appendix A). Monitoring of the geometry of S-H⋯S (**1**) showed a very low probability (0.1–0.9%) of such an interaction in all proteins. Contact (**2**) has a probability of 72% in Erp18 and 27% in Tmx1. Analysis of the contact metrics (distance S⋯S and angle at H-atom) indicated that a typical S-H⋯S H-bond is slightly stronger in Tmx1 than ERp18. Such an H-bond was not observed in the other studied proteins.

As expected, the S-H⋯S H-bond does not influence the folding of the CX1X2C motif (Appendix A). For instance, this H-bond is observed in conformations from clusters C1, C2 and C4 of ERp18, and it is absent in the others (C3 and C5), although the CX1X2C motif is well folded in both cases. Interestingly, both thiol groups do not contribute to H-bond interaction in the most prevalent PDI conformation with an unfolded CX1X2C motif, but K41, which is next to the C40 residue, is H-bound to H39 and L43. In the folded CX1X2C motif of PDI, C37 is in contact with P35 through the H-bond formed by the main chain atoms. Apparently, this interaction contributes to the stabilisation of the PDI conformation in the folded state, but it is not the unique factor that leads to such a structure. Similar but not-equivalent H-bonds are observed in the well-structured Tmx1 motif and the fully unfolded Tmx4 motif.

Structure organisation of the CX_1_X_2_C motif strongly influences their reactivity, affecting such properties as their accessibility and protonation state (i.e., p*K*_a_) [21]. Functional analyses of each cysteine in the consensus CX_1_X_2_C motif demonstrated that *N*-terminal cysteine is important for the formation of a transient S−S bond with the substrate, whereas *C*-terminal cysteine is involved in substrate release [22]. In proteins, specific hydrogen-bond donors and an electropositive local environment tend to lower the pK_a_ by stabilising thiolate, and a hydrophobic environment or an electronegative local environment tends to raise the pK_a_ by destabilising a negatively charged, as opposed to a neutral form, side chain [21,23,24].

### 2.2. Human VKORC1 Viewed as the Target of a Trx-Fold Protein

#### 2.2.1. General Characterisation

Human VKORC1 is composed of two domains: the extended luminal loop (L-loop), which contains the cysteine residues that participate in the electron exchange between the redox enzyme and hVKORC1, and the transmembrane domain (TMD), which includes two other cysteine amino acids from the highly conserved CXXC active site that is essential for vitamin K quinone reduction [25,26]. Based on studies of bacterial VKOR homologues, it was proposed that the loop cysteines of hVKORC1 allow protons to be shuttled to the active-site cysteines [12,27].

Earlier, a four-helix transmembrane domain structural model of human VKORC1 in its four functional states was reported [6]. Here, the focus is on the inactive (oxidised) state of hVKORC1, in which two pairs of cysteine residues, C43–C51 and C132–C135, are covalently linked to form disulphide bridges S⋯S (Figure 1D). hVKORC1 was studied by MD simulations of the model that mimics the protein in its natural environment, i.e., hVKORC1 embedded in the membrane and surrounded by water molecules (Figure 6A). While the extended L-loop (R37-N77 aas) has demonstrated high conformational variability in the protonated forms of hVKORC1 [6], the inactive state of hVKORC1 was studied by repeated 500 ns MD simulations (replicas **1**‒**3**) using random initial velocities.

The RMSDs computed for the positions of all Cα-atoms relative to the initial structure (*t* = 0 ns) showed comparable behaviour over the three MD trajectories, with a mean value (mv) of 5 Å (Figure 6B). The per-domain RMSDs showed that the *N*- and *C*-terminals are the fragments that contribute most to large RMSD values (up to 13 Å), while the TMD curves demonstrate a highly stable profile with the smallest RMSDs (2 Å). RMSDs computed for the Cα-atoms of the L-loop, after fitting to its initial conformation, showed alternated values, small or large, that were maintained over a large time scale (50–100 ns). The altered RMSD values, viewed as a set of well-defined slopes, indicate the possible conformational transitions in the L-loop. To check the suggested conformational transitions, MD conformations picked before and after each sudden RMSD change were compared. Three conformations of the L-loop that were chosen from replica **3** at *t* = 150, 250 and 375 ns showed significant differences in the folding and orientation of the helices and the loops, which revealed structural and conformational transitions (Figure 6E).

The larger RMSD values computed for Cα-atoms of the L-loop, after rigid alignment based on the initial conformation of the TMD compared to the RMSDs computed after rigid alignment based on the initial conformation of the L-loop, suggest the displacement of the L-loop from the TMD as a pseudorigid body (Figure 6C). The profile of the RMSF curves is similar in the three MD trajectories, with differences only in the amplitude of the RMS fluctuations of the highly flexible regions of hVKORC1, the *N*- and *C*-terminals and the extended L-loop (Figure 6D). The 2D and 3D structures of VKORC1 is generally conserved over the MD trajectories and shows a fully helical fold of the protein, with the four long-living extended (of 15–19 residues) transmembrane α-helices, TM1–TM4, observed in the reduced forms of hVKORC1 [6] and the three short helices on the L-loop (Figure 6E,F).

#### 2.2.2. The Luminal Loop of hVKORC1: Structure and Dynamics

Since the luminal loop (L-loop) is the fragment targeted by a Trx-fold protein, our focus is mainly on its intrinsic structural and dynamical properties and their connection with those of the transmembrane domain of hVKORC1.

L-loop folding, encompassing 30%, 36% and 22% of all residues in replicas **1**–**3,** respectively, is presented by three small (3–4 residues) transient helices, H1-L, H2-L and H3-L, which are partially converted between the αH- and 3_10_-helices (Figure 6E and Appendix A). Despite the transient structure of helices, their positions on the sequence are well conserved. The L-loop helices are interconnected by coiled linkers, which, together with the linker joining the L-loop to TM1 from the transmembrane domain of hVKORC1, display RMSF values that suggest the high mobility of these loops (Figure 6D). H1-L, mainly folded as a regular α-helix, contains C43 at its *C*-cap, which is linked covalently to C51, an *N*-cap residue of H2-L helix, forming the S⋯S bridge between two cysteines. Such covalent bonding significantly restricts the conformational mobility of this fragment. The large coiled linker connecting H2-L and H3-L helices is composed of hydrophobic residues, with the inserted charged and polar amino acids in the proximity of each helix (Figure 6F).

The intrinsic dynamics of hVKORC1 was first analysed with the cross-correlation matrix computed for the Cα-atom pairs of the full-length protein and the L-loop. The Cα–Cα distance pairwise patterns demonstrate the coupled motions within each hVKORC1 domain, the L-loop and the TMD and between two structural domains (Figure 7A and Appendix A). The regular pattern in the TMD reflects the correlated motion of the TM helices that is mainly associated with their collective drift, observed earlier in all metastable states of hVKORC1 [6]. The motion of the L-loop correlates with the movement of the linkers that connect the TM-helices and join the L-loop to the TMD.

The cross-correlations computed on only L-loop atoms display different maps in the three replicas, with either a fine-grained pattern (replicas **1** and **2**) or a pattern composed of well-defined blocks of nearly equal size (replica **3**), reflecting the highly coupled motion of the L-loop fragments consisting of 10–12 residues from the L-loop helices and their adjacent linkers. The difference in cross-correlation patterns is associated with the disparity of L-loop motion—small or medium in replicas **1** and **2** and broad in replica **3**, as evidenced by RMSFs and PCA.

The collective motions of VKORC1, characterised by PCA, showed that ten modes describe ~80–90% of the total fluctuations of both the full-length VKORC1 and the L-loop (Figure 7B). Similar to the RMSF values, the first two PCA modes denote the great mobility of the terminal residues (*N*- and *C*-terminus) and the L-loop (Figure 7B insert). PCA analysis performed on only the Cα-atoms of the L-loop showed that two first modes characterise most of the L-loop motion that displays the large-amplitude collective movements of the L-loop fragments—helices and adjacent-coiled linkers. The amplitude and direction of motion of the L-loop fragments differ in the three trajectories (Figure 7C), suggesting a larger conformational space for the L-loop than was observed in each trajectory, probably larger than the total space of all trajectories. The first two modes in replicas **1** and **2** showed a highly coupled motion of the H1-L helix and the L23 linker in a scissors-like manner, while the collective motion in **3** mainly displays a displacement of L23, which is horizontal with respect to the rest of the L-loop and vertical with respect to the TMD (Figure 7C and Appendix A).

To characterise the conformational changes of the L-loop that are associated with a great deal of flexibility and mobility, the most emblematic residues, in view of their fluctuations (RMSFs), were first selected. Two sets of residues—(1) C43, V54 and S74, located on the L-loop helices (the midpoint residues of H1-L, H2-L and H3-L) and showing the minimal values of RMSFs, and (2) R35, G46 and G64, positioned on the L-loop linkers L11, L12 and L13, respectively, and displaying the greatest RMSF values—were chosen (Figure 8A). Each set of residues was completed by residue C135 from the TMD and was then used to define two tetrahedrons, **T1** and **T2**, designed on the Cα-atoms. It is suggested that light may be shed on the conformational features of the L-loop by analysis of the six straight edges corresponding to the distances between each pair of residues.

Analysis of **T1** geometry showed (i) the great stability of Cα–Cα distances (d) between C43 (H1-L helix), V54 (H2-L helix) and C135 over nearly all the simulated time and in all the replicas; (ii) high conservation of Cα–Cα distances between each of the three residues and S74 (H3-L) over a substantial time period (200–300 ns or more), followed by (iii) a synchronic change of these distances (Δ of 6–8 Å), indicating the displacement of the H3-L helix with respect to the other helices, H1-L and H2-L (Figure 8B). As was expected, **T2,** which is determined using the most fluctuating residues, showed less conserved geometry, displaying synchronic changes in all or at least 3–4 distances (Δ of 8–15 Å).

Comparison of **T1** and **T2** metrics showed an absence of coupling between their geometries. Similarly, no evident relation was found between the secondary structure of the L-loop and the **T1** or **T2** geometries, suggesting that the relative positions of the residues from the L-loop helices and from the linkers connecting these helices are disconnected from the folding–unfolding effects in the L-loop.

This analysis revealed (i) the high stability of the H1-L helix in terms of its secondary structure, as well as in its relative position with respect to TMD, (ii) the quasistable spatial position of the transient H2-L helix relative to H1-L and TMD, (iii) the large displacement of the transient H3-L helix from the anchored structural motif formed by HL-1 and H2-L and of the coiled linkers L11–L13.

To illustrate the relative orientation of the L-loop helices, their structural drift was analysed. The axis of each helix was defined for the conformations from trajectories **1–3** (sampled every 100 ps, concatenated data), superposed and projected on a randomly chosen conformation of the L-loop (Figure 8D). The superimposed axes (elongated by 50% to better represent their position and direction) form a reap-like distribution for all helices. The axes of the three helices differ in length and their spatial orientation within each reap-like distribution and between the helices.

#### 2.2.3. Conformational Variability of the hVKORC1 L-Loop

To characterise the conformational space explored over the MD simulations using the L-loop of hVKORC1 in its inactive state and to distinguish the most probable conformations, the generated conformations were analysed using ensemble-based clustering [28]. Conformations of each MD trajectory were grouped with different RMSD cut-off values that varied from 1.6 to 3.0 Å, with a step of 0.2 Å. Using of cut-off value ≥2.2 Å results in a poor number of clusters, while more restricted cut-off values of 1.8 and 2 Å were sufficient to regroup the L-loop conformations into clusters that give the best cumulative population (>90%; Figure 9A). Interestingly, clustering with these cut-off values produces an equal number (six) of clusters, with a nonzero population in all replicas (Appendix A). Taking a cut-off of 2.0 Å as the criterion, the population of each cluster obtained for each trajectory was compared. In replica **1**, the majority of conformations form the two most-populated clusters, C1 (48%) and C2 (32%); the other conformations are regrouped in clusters with a low population of 0.5–9%. In each of the other replicas (**2**/**3**), the MD conformations are regrouped in the three most-populated clusters, with a comparable density between the replicas: C1 (41/33%), C2 (22/23%) and C3 (20/15%). The MD conformations that form the most populated clusters, C1 and C2, are individually regrouped within the narrow time ranges in trajectory **3** only, while in two other simulations, they are observed over a long period for each trajectory as coexisting with the conformations from the other clusters (Figure 9B). The conformations from the lowly populated clusters are usually observed in time ranges where the RMSD varies significantly and may show the transient states of the L-loop.

The representative conformations from different clusters of the same replica are divergent at the folding level (2D) and in 3D-structure organisation (Appendix A). An archetypical example is the considerable disparity between the conformations from clusters C2 and C3 of trajectory **3** that represents the L-loop before and after the transition, which is evidenced by the RMSD curve (Figure 6B). In contrast, some representative conformations of the clusters from different replicas showed a convenient similarity, for instance, C2 and C1 from replicas **1** and **3**, respectively.

It is supposed that the L-loop conformational spaces generated by the three independent trajectories are partially overlapped. To verify this hypothesis, a clustering analysis was performed on the merged trajectory composed of the L-loop conformations from three replicas. Accordingly, the number of clusters obtained with the same RMSD cut-off values is significantly lower for the merged data than the sum of clusters obtained individually for each replica (Figure 9A), which confirms the overlapping of the conformational spaces of the L-loop covered over the three replicas of MD simulation of hVKORC1 in its inactive state.

The first three clusters of the concatenated trajectory (cut-off 2.0 Å) contain 31%, 14% and 12% of all conformations, while the other conformations form the poorly populated clusters. The cumulative population of the clusters, with a density >4% on the merged data, is reduced (72%) with respect to individual trajectories but is still meaningful and statistically rich for the characterization of the most frequent L-loop conformations. Regarding the composition of the clusters, it was found that the dense clusters of the merged trajectory, C1^m^ and C3^m^, are composed of conformations from different trajectories (C1^m^ and C3^m^ are comprised of conformations from replicas **1**/**2**/**3**, with proportions of 83/4/12% and 28/12/58%, respectively), while the other clusters are composed of conformations from the unique trajectory—**2** (C2^m^ and C5^m^) and **3** (C4^m^ and C6^m^), respectively (Figure 9C).

The representative conformation of each cluster, generated for the concatenated trajectory, showed that the principal factors leading to the conformational difference of the L-loop consist of (i) a variable length of the H2-L helix; a decrease of that promotes (ii) an elongation of linker L23, which, in turn, encourages (iii) the repositioning of the H3-L helix with respect to the H1-L and H2-L helices (Figure 9C,D). In contrast to H2-L, the length of the H1-L and H3-L helices is better conserved. The whole shape of the conformations from different clusters well-reflects the “scissor-like” motion of the H1-L helix and the L23 loop that is observed in the PCA modes. The compact shape of the L-loop corresponds to the “closed” position of the H1-L helix and the L23 loop, which is a typical feature of most L-loop conformations (see the highly populated clusters, C1^m^–C4^m^). The conformations grouped in cluster C6^m^ show an elongated shape, with an “open” position of the H1-L helix and the L23 loop. Cluster C5^m^ is composed of intermediate conformations between the “open” and “closed” forms.

The clustering enabled (i) the splitting of MD conformations of the L-loop into groups composed of similar geometry and shape (within a cut-off), (ii) the assembly of a great majority of conformations into a limited number of clusters, and (iii) a distinction between dense clusters with a statistically reasonable population.

#### 2.2.4. Intra-L-Loop Interactions

To establish the forces that stabilise L-loop conformations, contact maps were computed for each representative conformation from the most populated clusters (>4%) found on the concatenated trajectory. The contact maps show the multiple intra-L-loop interactions between the linkers, between the linkers and helices and between the helices (Appendix A). Nevertheless, the patterns of such contacts differed in clusters C1^m^–C6^m^. The most common pattern found in the maps describes the contact of L11 with H2-L and H3-L helices and of L23 with H3-L, which are systematically observed in clusters C1^m^–C5^m^.

Analysis of the H-bonds showed that the L-loop conformations are stabilised by mutual H-bonds that form extensive networks (Figure 10, Appendix A). Comparing these H-bond networks in “closed” conformations (clusters C1^m^–C5^m^), it is noted that D36, D38, D44, R53, R61 and E67 are the key residues that form the salt bridges. In the “open” conformation (cluster C6^m^), the set of interacting residues that form the salt bridges is composed of R35, D38, D44 and R58.

The salt bridge that is stabilised by the pairing of charged residues when a combination of two noncovalent interactions is formed, H-bonding and ionic-bonding, is the most commonly observed contribution to the stability of the entropically unfavourable folded conformation of proteins [29]. Indeed, in the highly compact “closed” L-loop conformations from C1^m^ and C2^m^, R53 interacts with D36 and D38, forming the R53-based “salt bridge pattern” that stabilises the proximal position of H2-L and the L11 linker. In the conformations from cluster C3^m^, the “salt bridge pattern” is formed by R61 interacting with D36 and E67, which stabilises the tight location of two distant linkers, L11 and L23.

These interactions in C3^m^ are completed by the contact of R53 (H2-L) with D36 (L11), causing an overlap of the two “salt bridge patterns”, namely, R61- and R53-based patterns. Additionally, in C3^m^, the other “salt bridge pattern” is formed by R37 contacting with D44, which stabilises H1-L and the L12 loop in a tight spatial position. In C4^m^, the R53- and R61-based “salt bridge patterns” are clearly separated, while each positively charged residue interacts with different subsets of the negatively charged residues, i.e., R61 with D36 and E67 and R53 with D38. These two “salt bridge patterns” gather together two neighbouring helices, H1-L and H2-L, and two distant linkers, L11 and L23. In C4^m^, similar to C3^m^, the “salt bridge pattern” formed with R37 and D44 is clearly separated from the R61- and R53-based “salt bridge patterns”. Such a spatial separation of two “salt bridge patterns” is observed in the “open” conformations of the L-loop from C5^m^ and C6^m^ clusters, in which two “salt bridge patterns” are formed by R53 (H2-L) interacting with D36 and D38 (C5^m^) or with D38 and D44 (C6^m^), and either by R37 (L11) interacting with R40 and D44 (C5^m^) or by R35 bound to R35 and D38 (C6^m^).

Besides the salt-bridge interactions, the charged residues also contribute to H-bonds by interaction with the different polar and hydrophobic residues, which either act as H-donors or H-acceptors for the atoms in their main or side chains. All these ionic and H-bond interactions between the charged residues and between the charged and polar residues contribute to the tight spatial L-loop arrangement, in which the helices and linkers from the remote sequence segments are localised at close proximity. It is interesting to note that R40, D44, R53 and R61 interact in any conformation of the L-loop, independent of the L-loop’s shape, by forming either the salt bridges or the H-bonds.

Nevertheless, many charged and polar residues that are not involved or are partially involved in intra-L-loop interactions protrude from the L-loop, as illustrated by the “closed” and “open” conformations of the L-loop (Figure 10B). Considering the spatial position of the solvent-exposed residues with respect to L-loop cysteine residues (C43 and C51) that participate in the thiol–disulphide exchange reaction, the residues from sequence S52 to E67 are most likely involved in interactions with a redox protein.

As the L-loop also contains a large number of hydrophobic residues, their contribution to intra- and intermolecular interactions was evaluated. Although hydrophobic forces are known to be relatively weak interactions, such interactions can add up to make an important contribution to the overall stability of a conformer or molecular complex [30].

Multiple contacts between the A41, G46, A48, I49, V54, L70 and L76 hydrophobic residues were observed in “closed” conformations, while in the “open” conformations, such contacts involved V45, F55, F63, L70 and L76 (Appendix A, Appendix A). These hydrophobic contacts may reflect the stabilising interactions that complete the *H*-bond contribution as well as the repulsive forces that equilibrate the strong salt-bridge interactions.

Similar to the charged and polar residues, some hydrophobic side chains are oriented toward the exterior of the L-loop, putting them in positions accessible to the solvent, such that the number of such residues is significantly higher in the “closed” conformations than in the “open” ones (Figure 10C, Appendix A**)**. One part of these residues (F55, G56, F63, L65, V66) belongs to the sequence S52–E67, which was postulated to be involved in interactions with a redox protein.

### 2.3. Modelling of Molecular Complex Formed by hVKORC1 and Its Redox Partners

The molecular complex of hVKORC1 was constructed with PDI to probe our hypothesis on the identification of a hVKORC1 redox partner (see the Discussion section); 3D models of the complex were constructed using the crystallographic structure of the VKOR from bacteria (bVKOR; PDB ID: 4nv5); [14]) as a reference for the initial positioning of PDI relative to hVKORC1.

To be most objective in the modelling of the human PDI–VKORC1 complex, the structure of bVKOR was not used as a template because of (i) the suggested alternative VKOR activation mechanisms in bacteria and in eukaryotes, that is, in their respective native environments, which employ significantly different mechanisms for electron transfer [14], (ii) a high structural difference between the Trx- and L-loop domains in bVKOR and human proteins (RMSD values are 4.5 and 4 Å between bVKOR and the “closed” and “open” conformations of hVKORC1, respectively), (iii) very low sequence identity/similarity (15/20%), and (iv) a very large distance between the cysteine residues from the Trx-like and VKORC1-like domains (the minimal S⋯S distance of 16 Å) in bVKORC1 (Appendix A).

For modelling the human PDI–VKORC1 complex, a conformation of hVKORC1 with the most extended “open” L-loop (the least probable conformation) was chosen as the initial target structure in order to bring the two proteins as close as possible. As for the initial PDI model, the conformation with a well-ordered and long αH2-helix that is similar to the X-ray structure of PDI [9] was chosen and positioned above hVKORC1 so that (i) the distance between the sulphur atoms from C37 of PDI and C43 of hVKOR1 was as short as possible (12.5 Å) and (ii) each PDI fragment that was suggested to be a fragment able to form the intermolecular interactions with a target, namely, **F1** and **F2**, was alternatively placed above the middle of the L-loop surface. The obtained promodels, **Model 1** and **Model 2**, were explored using MD simulation for conditions (see the Methods section), where restraints apply to the distance S⋯S between C37 (PDI) and C43 (hVKORC1). The restraints were gradually diminished during a stepped 80-ns MD simulation run (Figure 11).

For both models, structural rearrangement occurred inside each protein and between the proteins, with diminishing S⋯S distance. In **Model 1**, the extended “open” conformation of the hVKORC1 L-loop was observed at an S⋯S distance of 12 Å, which then adopted a “closed” conformation at a shortened S⋯S distance (of 10 and 8 Å), with the αH1-L helix and the L23 linker located in a proximal position, which is the most probable conformation of the L-loop in isolated VKORC1. The initially well-ordered and long αH2-helix of PDI is rotated by 30° (at an S⋯S distance of 10 Å), followed by the bending of the helix, and then (at the S⋯S distance of 8 Å) by depletion of two helices, a small 3_10_-helix in the proximity of the CX_1_X_2_C motif and a shortened αH-helix, which demonstrates a folding–unfolding effect observed in MD simulations of PDI in an isolated state.

Similarly, in **Model 2**, a gradually diminishing S⋯S distance from 12 to 8 Å promotes a change in the L-loop conformation from “open” to “closed” in hVKORC1, while in PDI, a departure of the αH3-helix from its initial position to the location most exposed to the solvent (a 4.5–5.0 Å parallel displacement of the helix) was observed. The conformational changes observed during the simulations of the two PDI–hVKORC1 complex models are reflected in the folding of “interacting” proteins. The extended “open” conformation of the hVKORC1 L-loop, taken as the initial structure for complex modelling, showed increased folding (by 50%) in **Model 1**, with a decrease in the S⋯S distance from 12 to 8 Å, while in **Model 2**, its helical fold was reduced by 40% (Appendix A). As for PDI, the folded content of its initial and final conformations was the same for both models.

Analysis of the intermolecular contacts at the interface between PDI and hVKORC1 (in the conformation taken at *t* = 80 ns) showed that these two proteins in **Model 1** are linked through two salt bridges formed by R61 (hVKORC1) and E46 (PDI) and by D67 (hVKORC1) and K49 (PDI) (Figure 12A). Hydrophobic contacts were also observed between two pairs of residues: A42 (PDI) and G62 (hVKORC1) and P45 (PDI) and L65 (VKORC1). Moreover, G62 (hVKORC1) interacts with P45 (PDI). The PDI–hVKORC1 interface interactions are completed by an H-bond between the side chain of S57 (hVKORC1) and the main chain of G38 (PDI), the amino acid in the proximity of the CX1X2C motif, and by the hydrophobic interaction between V45 (hVKORC1) and G82 (PDI). All distances between the interacting D⋯A atoms were ranged from 2.5 to 3.2 Å, which characterise strong interactions.

Spatially, two sets of interactions stabilising the PDI–hVKORC1 complex were observed. The first set, which is composed of S57(hVKORC1)⋯G38(PDI) and V45(hVKORC1)⋯G82(PDI), is localised in the proximity of the active sites, the CGHC motif of PDI and disulphide bridge C43–C51 of hVKORC1 and probably stabilises their close location, which is induced, in part, by a steric requirement imposed on the sulphur atoms from C37 and C43 to be in the closed position. The second set, which is composed of multiple contacts between the residues from short sequence segments, A42–K49 from PDI and R61–E67 from hVKORC1, forms a very compact regular interaction pattern that describes the highly specific recognition between two molecules that are maintained by two salt bridges and by crosswise hydrophobic interactions. This pattern of interactions stabilises the αH2-helix of PDI and the L23 linker from hVKORC1 in a close position that is independent of any interaction with Set 1 and, consequently, may present a first step in the PDI–hVKORC1 recognition process.

The residues of hVKORC1 that form salt-bridges and H-bonds are located on the transient H2-L helix and on the L23 linker, which is composed of a segment that was predicted to be the most putative recognition region in an isolated hVKORC1. Similarly, PDI residues participating in hVKORC1 recognition belong to the **F1** fragment were regarded as a possible putative recognition site. Surprisingly, a hydrophobic interaction with V45 (hVKORC1) is formed by G82 (PDI), which is a residue from the **F2** fragment that is also predicted to be a fragment that contains possible recognition sites.

In **Model 2**, the interaction interface between PDI and hVROR1 is also formed by two salt bridges generated by R35 (hVKORC1) and D67 (PDI) and by E67 (VKORC1) interacting with R81 and K87 from PDI. Other electrostatic interactions are presented by the H-bonds of Q72 (hVKORC1) with A68 and S72 from PDI and of R61 (hVKORC1) with V80 of PDI. Hydrophobic contacts are observed as a three-furcate interaction of the three PDI amino acids (A75, G79 and V80) attached to a unique amino acid (V69) of hVKORC1. Unlike the compact interface contact network in **Model 1**, the interacting residues in both proteins of **Model 2** are distributed over large sequence segments, from D67 to K87 in PDI and from R35 to Q75 in hVKORC1. This highly enlarged interface interaction network seems less probable because of the small probability of a synchronised approach of two space-separated binding sites to the target.

It is interesting that two amino acids, R61 and E67, of hVKORC1 form salt bridges in both models, **Model 1** and **Model 2**, but by selecting different PDI residues. Remarkably, both amino acids belong to a hVKORC1 segment that is predicted to be the putative recognition site by analysis of the isolated protein.

Although very compact and regular, the interface interaction pattern formed by the closely localised residues in both proteins from **Model 1**, together with the increased helical folding of the L-loop by 50%, is a very attractive argument for the choice of this model for being functionally related, though there is still doubt in such a conclusion.

Other characteristics are considered to better justify or challenge our hypothesis. First, from a superimposition of each model on the experimentally defined structure of bVKORC1, the best fit at the level of Trx-like domain orientation with respect to VKOR is observed for **Model 1** (Appendix A), but analysis of the interaction between the Trx-like and VKOR domains in the bacterial protein showed only a single short contact (between Q40 from the αH2-helix of Trx and L46 of the L-loop), an observation that largely mismatches the interaction patterns observed in both models.

Finally, to check the stability of the interactions between the two proteins in **Model 1** and **Model 2**, the models were simulated at *t* = 80 ns under more relaxed (“soft”) conditions (see the Methods section), which gives more tolerant restrains on the distance S⋯S between C37 (PDI) and C43 (hVKORC1).

In the two MD simulations of **Model 1**, which have different “soft” constraints (a time range of 80–100 ns), the distance S⋯S either varied within an enlarged range (7–11 Å) or, surprisingly, showed a tendency to decrease (6–10 Å) with respect to the simulation with a more “hard” restriction (a time range of 60–80 ns) (Appendix A). The MD conformations of **Model 1**, generated using different “soft” constraints, showed very similar structures of PDI–hVKORC1 that differed only in the folding of the H2-L helix from the L-loop of hVKORC1 and the αH2 helix of PDI. Each of these structural effects was observed in isolated proteins. The interface interactions between the residues from the αH2 helix of PDI and L23 from the L-loop of hVKORC1 were very similar for conformations taken at *t* = 100 ns and *t* = 80 ns. With respect to the conformation chosen at *t* = 80 ns, some novel contacts involving residues from H2-L (hVKORC1) and the L3 loop and of PDI are observed in the conformation taken at *t* = 100 ns (Appendix A).

These results show that the highly specific recognition between the two molecules is maintained by the strong and stable interactions formed by two salt bridges and by crosswise hydrophobic interactions preserved in **Model 1**.

The MD conformations of **Model 2**, generated using “soft” and “hard” constraints, showed similar structures of PDI–hVKORC1, which differed only in the position of the αH3 helix of PDI and the L-loop of hVKORC1. The interactions observed at the interface between the two proteins are nonpreserved, with the exception of a single salt bridge between E67 (hVKORC1) and R81 (PDI) (Appendix A).

## 3. Discussion

Vitamin K epoxide reductase is a membrane protein that reduces vitamin K using a membrane-embedded cysteine-containing redox centre. Such activity requires the cooperation of VKORC1, with a redox partner that delivers reducing equivalents. The physiological redox partner of hVKORC1 remains uncertain; nevertheless, four proteins—PDI, ERp18, Tmx1 and Tmx4—were suggested as the most likely *H*-donors of hVKORC1 [12,13]. Deciphering the molecular origins of VKORC1 recognition by an unknown redox protein is not a trivial task.

We suggested that a careful in-silico study of the isolated proteins would provide useful information. In particular, quantitative metrics and qualitative estimations can shed new light on the target (hVKORC1) features and the peculiarities of redox proteins. Such information may help in predicting (i) the protein fragments participating in VKORC1 recognition by a Trx and (ii) the most probable partner of VKORC1.

What has been learnt from studying VKORC1 and four Trx-fold proteins?

The L- loop is known to bind to and accept reducing equivalents from species-specific partner oxidoreductases essential for VKOR enzymatic function in vivo [31], so this domain was carefully characterised. We found that the L-loop in the inactive (oxidised) state of VKORC1 is noticeably less flexible compared to the reduced states of VKORC1 [6] and more folded, showing three helices connected by coiled linkers. This three-helix fold of the L-loop was generally maintained over the MD simulations, while the length and spatial positions of the helices were highly variable. This variation is reflected in a large number of L-loop conformations, varying from a compact “closed” conformation, which is prevalent, to an extended “open” conformation.

It was established that the H2-L helix is the fundamental actor that controls the conformational features of the L-loop. This transient helix converts between the αH- and the 3_10_-folds, adapting in length from short to elongated. The shortened H2-L helix, in which the S56-R61 segment is unfolded, promotes the elongation of the coiled linker L23, connecting the H2-L and H3-L helices. The extended linker L23 shows (i) great mobility with respect to the H1-L helix, which can be described as a “scissor-like” motion, and (ii) a large vertical displacement with respect to the TMD. Moreover, the extended linker L23 delivers increasing mobility to H3-L, evidenced by its displacement with respect to H2-L.

At the sequence level, the L-loop has been reported to be conserved between VKORs from different species [32]. Particularly high conservation was found for the S56–G63 segment, which in hVKORC1 is followed by 5-residue hydrophobic insert GFGLV, which is completed by glutamic acid (E67) and histidine (H68). Sequence conservation, along with observed structural and dynamical properties of the H2-L helix and its adjacent linker L23, suggests their possible functional role. From the analysis of the H-bonding patterns in the L-loop, regular exposition of the charged (R58, R61, E67 and D73) and polar residues (S56, S57, W59, H68 and N77) to the outer side of the L-loop was observed in positions favourable for contact with a solvent or protein. Therefore, we postulated that the S56–R61 segment, a part of the more extended S53–N77 segment, is a platform for the recognition of a protein partner.

Charged residues have been shown to be instrumental in the definition of binding specificity, while sometimes contributing little binding energy to the interactions themselves [33,34]. In other cases, charged residues were found to promote high-affinity binding [35,36]. They are also the main players in “electrostatic steering”, which is a long-range mechanism in which electrostatic forces can steer a ligand protein to a binding site on the receptor protein; this drastically increases the association rate [37,38]. Often, charged residues that are important for protein–protein interactions are conserved across families of evolutionarily related proteins and protein complexes [39,40,41].

Moreover, the tryptophan residue (W59) from the S56–R61 segment, following 5-residue hydrophobic insert GFGLV, may act as an anchoring residue that binds the two proteins. Tryptophan residues have been shown to exhibit a strong tendency to remain within the interfacial region [42]. The role of the hydrophobic effect as a driving force in protein folding and assembly is well described [43].

Which of the four studied Trx-fold proteins is the most probable partner for VKORC1?

Regarding the probable redox partners of hVKORC1 (Erp18, PDI, Tmx1 and Tmx4), it was observed that despite their similar architecture, each protein is characterised by its own sequence-dependent structural and dynamical features. In particular, it was observed that the CX_1_X_2_C motif’s different folds are connected to the divergent configuration of the thiol groups—either as part of the well-folded αH2-helix (Erp18 and Tmx1), with the restrained cis-geometry of sulphur atoms, or as a part of a coiled structure, with the alternating orientation of sulphur atoms that runs from a syn-periplanar configuration to an antiperiplanar configuration (Tmx4), or as part of a transient structure (PDI) reversed between the helical fold (α- and 3_10_-helices) and turn-coil structure, leading to a large number of thiol group configurations.

Focusing on the **F1** region, suggested to be able to form intermolecular interactions with a target, it is noted that only **F1** of PDI and the targeted S56–R61 segment of VKORC1 have similar structural properties, or rather, a structural disorder that describes an intrinsically disordered region (IDR). Indeed, two IDRs, which are the transient *N*-terminal of the α-helix H2 in PDI and the transient H2-L helix comprising the S56–R61 segment from the L-loop of VKORC1, show similar structural heterogeneity and plasticity that is consistent with an affinity that is sensitive to changes in local frustration distribution and thermodynamics.

Numerous publications have reported that many protein–protein interactions (PPIs) are mediated by protein regions that are not confined to a single folded conformation prior to binding, namely, IDRs that participate in PPIs (interacting IDRs) [44,45,46]. IDRs are increasingly recognised for their prevalence and their critical roles in regulatory intermolecular interactions [47]. It has been hypothesised that some traits make IDRs particularly suitable for interactions that involve signalling and regulation, complementing globular domains that more often perform catalytic functions. It has been estimated that IDRs in the human proteome contain ~132,000 binding motifs [48]. Disordered proteins are believed to account for a large fraction of all cellular proteins, playing roles in cell-cycle control, signal transduction, transcriptional and translational regulation, and large macromolecular complexes [49]. Nevertheless, even if fragment **F1** is considered the most probable fragment to form intermolecular interactions with a target, the mobility of linker L5 and the αH3 helix from **F2** of PDI means that **F2** has strong compatibility with the highly mobile S56–R61 segment of VKORC1. Moreover, **F2** shows the most dissimilar sequence in the studied proteins, and it also has a great number of hydrophobic, polar and charged residues that are exposed to solvents. Consequently, **F2** is also potentially able to contribute to stabilising a supramolecular complex.

These two fragments are very close to the CX_1_X_2_C motif, which is either joined in a sequence (**F1**; sequence vicinity) or adjacent in a 3D structural space (**F2**; spatial vicinity).

This makes clear that we can begin to construct models of the molecular complex formed by hVKORC1 and PDI, where PDI is the most probable redox partner of hVKORC1. Exploring the recognition processes between these two proteins, hVKORC1 and PDI, requires knowledge of the 3D structure of the associated molecular complex.

Direct use of the X-ray structure of VKOR from bacteria (a protein with covalently bound Tmx-like and VKOR-like domains, which has low sequence and structure similarity compared to the human proteins PDI and VKORC1) is not appropriate for the modelling of the human complex but can be a reference for the initial positioning of PDI with respect to hVRORC1. Using conventional MD simulations, two models of the PDI–hVKORC1 complex, with the PDI in two alternative positions, which were either exposed by **F1** (**Model 1**) or **F2** (**Model 2**) in front of the L-loop of hVKORC1, were studied. In both probed models, proteins bind to each other using a combination of hydrogen bonds, salt bridges and hydrophobic contacts formed by residues from the different protein domains. These domains are small binding clefts and include a few peptides in **Model 1**, while in **Model 2**, the molecular interface represents large areas on each protein and spans widely spaced amino acids in protein sequences.

How do the “interacting” residues predicted by analysis of isolated proteins correspond to the contacts in the complex formed by VKORC1 and PDI?

In **Model 1,** the interface contact network is composed of two salt bridges formed by two pairs of charged residues, R61 and E67, from hVKORC1, which, together with S57, G62 and L65, also contribute to the stabilisation of the two proteins. These residues are amino acids from the L-loop segment that was predicted as a platform for recognition of a protein partner by hVKORC1. In **Model 2,** the interaction interface between PDI and hVROR1 is also completed by two salt bridges formed by R35 and E67 from hVKORC1 interacting with D67, R81 and K87 from PDI and by *H*-bonds formed by Q72 and R61 of hVKORC1 with A68, S72 and V80 of PDI. In both models, two amino acids of hVKORC1, R61 and E67, participate in strong electrostatic interactions, salt bridges or *H*-bonds but with different PDI residues. It is remarkable that both amino acids belong to a hVKORC1 segment that is predicted to be the putative recognition site from analysis of the isolated protein. The contacting PDI residues are mainly predetermined by PDI orientation with respect to the L-loop.

Based on limited data from the stepped finite-time simulations, is it possible to conclude which model is the correct one?

In both models, the optimised (enhanced) orientation of PDI with respect to hVKORC1 is maintained by the multiple interactions between the two molecules.

In **Model 1**, intermolecular contacts are observed between the two short length peptides, R61-E67 from hVKORC1 and A42–K49 from PDI, which form two salt bridges and three crosswise hydrophobic interactions. Such a compact regular interaction pattern may describe highly specific recognition between the two molecules, maintaining the αH2-helix of PDI and the extended L23 linker of VKORC1 in a close position and, consequently, may present the first step in the PDI–hVKORC1 recognition process. The other set of interactions, S57(hVKORC1)⋯G38(PDI) and V45(hVKORC1)⋯G82(PDI), is located in close vicinity to the CGHC motif of PDI and disulphide bridge C43-C51 of hVKORC1. This is induced by a steric requirement imposed on the sulphur atoms from C37 and C43 that holds them in a closed position. Moreover, as these contacts are formed by the main chain atoms, they are rather nonspecific.

In **Model 2**, the interaction interface between PDI and hVROR1 represents a large area for each protein and spans long-spaced amino acids of the protein sequences (D67–K87 in PDI and R61–Q72, completed by R35 in hVKORC1). The two salt bridges, which are formed by R35 (L11 from hVKORC1) and D67 (L5 from PDI) and by E67 (L23 from VKORC1) interacting with R81 and K87 from L6 of PDI, involve two regions on each protein that are separated by large distances in the sequence and the 3D structure. The other *H*-bonds involve the residues located between the two remote salt bridges. The dense cluster of hydrophobic contacts is realised as a three-furcate interaction of three PDI amino acids (A75, G79, and V80) attached to a single amino acid (V69) of hVKORC1.

In both models of the PDI–hVKORC1 complex, interacting hydrophobic motifs from both proteins form “interacting hydrophobic cores”, which may be the key factors in the recognition process. The total number of noncovalent contacts between PDI and hVKORC1 in **Model 2** is 9, while in **Model 1**, it is only 5. It was reported that the number of connections between each pair of proteins is a strong predictor of how tightly the proteins connect to each other [50].

Nevertheless, despite the large number of *H*-bonds and the dense cluster of hydrophobic contacts, it appears that the enlarged interface interaction network observed in **Model 2** is less likely, due to the low probability of a synchronised approach of the two space-separated binding sites on PDI to the two space-separated binding sites on the target.

Moreover, based on the stepped simulations of **Model 1**, the diminishing distance between the two proteins promoted an increase in the helical folding of the L-loop by 50%, while in **Model 2**, its helical fold was reduced by 40%. While proteins become disordered on their own, their native conformation is stabilised upon binding [51,52]. The folded content of the initial and final PDI conformations is the same in both models; nevertheless, its conformation is adapted in both models by the folding–unfolding of the αH2-helix in **Model 1** and by the removal of the αH3 helix in **Model 2**.

The specificity of intermolecular interactions in PDI–hVKORC1 is apparently determined by sequence- and structure-based selectivity, which are the two determining factors in “molecular recognition”. A natural implication of the conformational selection model is the particular range of surface shapes visited by each protein and their collective complementarity, which is adjusted throughout the binding process. It was recognised that cooperativity derives from the hydrophobic effect, the driving force in single-chain protein folding [53]. The hydrophobic folding units that are observed at the interfaces of two-state complexes similarly suggest the cooperative nature of two-chain protein folding, which is also the outcome of the hydrophobic effect [54,55,56]. Nevertheless, although the hydrophobic effect plays a dominant role in protein–protein binding, it is not as strong as that observed in the interior of protein monomers; its extent is variable. The binding site is not necessarily at the largest patch of the hydrophobic surface. There are high portions of buried charged and polar residues at the interface, suggesting that hydrogen bonds and ion pairs contribute more to the stability of protein-binding than to that of protein-folding. Protein-binding sites have neither the largest total buried surface area nor the most extensive nonpolar buried surface area. They cannot be uniquely distinguished by their electrostatic characteristics, as observed by parameters such as unsatisfied buried charges or the number of hydrogen bonds.

The question is then to test if electrostatic and hydrophobic interactions in the PDI–hVKORC1 complex can be conserved qualitatively. The MD simulations of **Model 1**, performed upon different “soft” constraints that supplied an increased degree of freedom for proteins and allowed them to be removed, proved the stability of the interactions formed by salt bridges and by the crosswise hydrophobic contacts. As **Model 1** of the PDI–hVKORC1 complex showed stable interface interactions under such conditions, it was proposed as the first precursor to probe thiol–disulphide exchange reactions between PDI and hVKORC1.

Returning to the questions stated at the beginning of this work, they seem to have all been answered using a purely in-silico approach. Molecular modelling and molecular dynamics simulations provide powerful tools for the exploration of proteins and their complexes. Such a study is most effective when analysed in close conjunction with experiments on a protein function, which would play an essential role in validating and improving the modelling and simulations. Therefore, we are now waiting for needed experimental validation (currently being undertaken by biologist colleagues) of the predictions given in this article. Experimental validation of the model of the PDI–hVKORC1 complex is essential for the continuation of this research, which will allow a better understanding of the redox chemistry underlying vital cell processes.

## 4. Materials and Methods

### 4.1. 3D Models

**Trx-fold proteins.** Structures of PDI (PDB ID: 4ekz), ERp18 (PDB ID: 1sen) and TMX1 (PDB ID: 1x5e) were retrieved from the PDB database [5] and atomic coordinates of domain **a**, which contains the CX_1_X_2_C motif and is present in all available structures, were extracted. The 3D homology model of hVKORC1 was generated from human sequence Q9H1E5 (https://www.uniprot.org/uniprot/) using the Modeller program [57] and the empirical structure of TMX1 (PDP ID: 1x5e) that was used as a template. The 3D model of the h-ERp18 protein was optimised (the cysteine residues were saturated with hydrogen atoms) to obtain a reduced state of the CX_1_X_2_C motif.

**h-VKORC1**. The coordinates of full-length hVKORC1 (sequence M1–H163) in the inactive state was taken from [6].

**Trx-VKORC1 complex**. Each complex of the PDI protein with hVKORC1 (PDI–hVKORC1) was modelled using the structure of bacterial VKOR (bVKOR; PDB ID: 4nv5) as a reference for the initial PDI positioning with respect to hVKORC1. The structures of the human Trx-fold protein and hVKORC1 were carefully superimposed with the respective domains of bVKOR. To eliminate a small intersection between part of the L-loop of hVKORC1 and PDI, the extended conformation of the L-loop was chosen. The PDI protein was placed in two orientations with respect to VKORC1, with (i) L3 and the αH2-helix (**F1**) and (ii) L5 and the αH2-helix (**F2**) positioned in front of the predicted “binding fragment” of the L-loop from hVKORC1. The initial distance S⋯S between the sulphur atoms from C37 of PDI and C43 of hVKORC1 in each built complex was 16 Å.

The stereochemical quality of all 3D models was assessed by Procheck [58], which revealed that more than 95% of nonglycine/nonproline residues have dihedral angles in the most favoured and permitted regions of the Ramachandran plot, as is expected for good models.

### 4.2. Molecular Dynamics Simulation

#### 4.2.1. Preparation of the Systems

For MD simulations, all models of the isolated proteins (PDI, ERp18, Tmx1, Tmx4), hVKORC1, and the two models of the PDI–hVKORC1 complex in two orientations (PDI**_F1_**–VKORC1 and PDI**_F2_**–VKORC1) were prepared with the LEAP module of Assisted Model Building with Energy Refinement (AMBER) [59] using the ff14SB all-atom force field parameter set [60]: (i) hydrogen atoms were added; (ii) covalent bond orders were assigned; (iii) protonation states of amino acids were assigned based on their solution for pK values at neutral pH and histidine residues were considered neutral and were protonated for ε-nitrogen atoms; (vi) the Na^+^ counter-ion was added to neutralise the protein charge.

Each membrane protein, hVKORC1 and the two models of complex PDI–VKORC1 (PDI**_F1_**–VKORC1 and PDI**_F2_**–VKORC1) were embedded in the equilibrated and hydrated membrane composed of 200 1,2-dilauroyl-sn-glycero-3-phosphocho-line (DLPC) lipids using the replacement method available in the CHARMM-GUI membrane builder (http://www.charmm-gui.org/input/membrane) [61]. This lipid bilayer was completed with 17293 (hVKORC1), 22047 (PDI**_F1_**–VKORC1) and 22567 (PDI**_F2_**–VKORC1) water molecules (TIP3P; [62]), pre-equilibrated during 1.5 ns of MD using the *Lipid14* tool [63] from the AMBER package.

Each protein or protein complex inserted into a membrane was solvated with explicit TIP3P water molecules in a periodic rectangular box with a distance of at least 12 Å between the proteins and the boundary of the water box. Cl^–^ ions were randomly placed to neutralise the system.

The total number of atoms in the isolated Trx-fold proteins (protein, water molecules and counter ion) varied from 16,065–26,386. The total number of atoms in the membrane systems (hVKORC1 and its complexes with PDI, including proteins, DLPC lipids, water molecules and counter-ions, was 72683 (hVKORC1), 92570 (PDI**_F1_**-VKORC1), and 93325 (PDI**_F2_**–VKORC1). The box size varied in the range of 84 × 84 × 108–141 Å^3^.

#### 4.2.2. Set-Up of the Systems

The set-up of the systems was performed with the Simulated Annealing with NMR-Derived Energy Restraints (SANDER) module [64] of AMBER18. First, each system was minimised successively using the steepest descent and conjugate gradient algorithms, as follows: (i) 10,000 minimisation steps where water molecules have fixed, (ii) 10,000 minimisation steps where the protein backbone is fixed to allow protein side-chains to relax, and (iii) 10,000 minimisation steps without any constraint on the system. The equilibration was performed on the solvent, keeping the solute atoms (except *H*-atoms) restrained for 100 ps at 310 K and a constant volume (NVT). Protein, membrane and solvent (water and ions) temperatures were separately coupled to the velocity rescale thermostat, which was a modified Berendsen thermostat [65] with a relaxation time of 0.1 ps. Each system was equilibrated for 1 ns (NPT), with all nonhydrogen atoms of the protein and the DLPC membrane harmonically restrained. Semi-isotropic coordinate scaling and Parrinello–Rahman pressure coupling were used to maintain the pressure at 1 bar, with a relaxation time of 5 ps. The Nose–Hoover thermostat [66] was applied to the protein, lipids and solvent (water and ions) separately, with a relaxation time of 0.5 ps to keep the temperature constant at 310 K. Water and ions were allowed to move freely during equilibration.

#### 4.2.3. Production of the MD Trajectories

All trajectories were performed using the AMBER ff14SB force field with the PMEMD module of AMBER 16 and AMBER 18 [59] (GPU-accelerated versions) running on a local hybrid server (Ubuntu, LTS 14.04, 252 GB RAM, 2x CPU Intel Xeon E5-2680 and Nvidia GTX 780ti) and the supercomputer JEAN ZAY at IDRIS.

The 500-ns MD trajectories of each fully relaxed isolated protein were generated (2 replicas for Trx-fold proteins and 3 replicas for hVRORC1) in its natural environment—the water solution for the Trx-fold protein and the solvated bilayer lipid membrane for h-VKORC1. Each PDI–hVKORC1 complex that was inserted into the solvated bilayer lipid membrane was simulated for an alternating value of distance S⋯S from PDI and hVKORC1 (see the next subsection for details). MD simulation of the Trx–VKORC1 complex was first performed for 38 ns, with a constrained S⋯S distance of 12.8 Å, which was further reduced to 10.2 Å and followed by simulation for 20 ns, and finally to 8.2 Å, followed by the last 20-ns of the simulation.

A time step of 2 fs was used to integrate the equations of motion based on the Leap-Frog algorithm [67]. Coordinate files were recorded every 1 ps. Neighbour searching was performed by the Verlet algorithm [68]. The Particle Mesh Ewald (PME) method [69], with a cutoff of 9.0 Å, was used to treat long-range electrostatic interactions at every time step. The van der Waals interactions were modelled using a 6–12 Lennard–Jones potential. The initial velocities were reassigned according to the Maxwell–Boltzmann distribution.

#### 4.2.4. The Stepped MD Simulations of the PDI–hVKORC1 Complex

In **Model 1** and **Model 2**, to prevent the separation of the PDI protein from hVKORC1 and to bring them together, a restrained harmonic distance was introduced to the S⋯S atom pair (the sulfur atoms from C37 of PDI and C43 of hVKORC1), which was varied in a stepwise manner (see Figure 7A). Specifically, the 80-ns simulation was divided into three steps, each with different applied restraints (*d*): from 0 to 38 ns with *d* equal to 12.8–11.8 Å (Step A), from 38 to 60 ns with *d* equal to 10.2–9.6 Å (Step B), and 60 to 80 ns with *d* equal to 9.2–8.2 Å (Step C). To probe the stability of the PDI–hVKORC1 complex, the simulations of **Model 1** and **Model 2**) were continued from 80 to 100 ns with two different “soft” restraints applied to distance S⋯S (see Appendix A). While the lower limit value remained at 8.2 Å, as in the previous simulation steps (A–C), the upper limit in Step D was increased to 10.2 Å (as in the 60–80 ns step) and 12.8 Å (as in the 0–38 ns step).

### 4.3. Data Analysis

#### 4.3.1. Conventional Analysis of the MD Trajectories

Unless otherwise stated, all recorded MD trajectories were analysed (RMSFs, RMSDs, DSSP, clustering) with the standard routines of the CPPTRAJ 4.15.0 program [70] of AMBER 18 Suite.

(1)The RMSD and RMSF values were calculated for the Cα-atoms using the initial model (at *t* = 0 ns) as a reference. All analysis was performed on the MD conformations (every 10 ps) by considering either all simulations or the production part of the simulation, which was generated after the removal of non-well-equilibrated conformations (0–70 ns), as was shown by the RMSDs, or on residues with a fluctuation of less than 4 Å, as shown by the RMSFs. For hVKORC1, the RMSDs were individually calculated for each domain after least-square fittings of the MD conformations to the initial conformation of a domain, thus removing rigid-body motion from the analysis.(2)Secondary structural propensities for all residues were calculated using the Define Secondary Structure of Proteins (DSSP) method [71]. The secondary structure types were assigned for residues based on backbone -NH and -CO atom positions. Secondary structures were assigned every 10 and 20 ps for the individual and concatenated trajectories, respectively.(3)The dynamic cross-correlation (DCC) between all atoms within a molecule quantifies the correlation coefficients of motions between atoms, i.e., the degree to which the atoms move together [72]. Calculations were performed on backbone Cα-atoms on the productive simulation time of each MD trajectory using an ensemble-based approach [28]. The correlation values vary between −1 and 1, where 1 illustrates a complete correlation, −1 a complete anti-correlation and 0 no correlation.

(4)The collective motions of proteins were investigated by principal component analysis (PCA). For an *N*-atom system, a trajectory matrix contains, in each column, Cartesian coordinates for a given atom at each time step $\left(x\left(t\right)\right)$
. Fitting the coordinate data to a reference structure results in the proper trajectory matrix $\left(X\right)$. The trajectory data is then used to generate a covariance matrix $\left(C\right)$, elements of which are defined as in Equation (1)
(1)$$c_{ij}=\langle \left(x_{i}-\langle x_{i}\rangle \right)\left(x_{j}-\langle x_{j}\rangle \right)\rangle$$
where $\langle \left(x_{i}-\langle x_{i}\rangle \right)\left(x_{j}-\langle x_{j}\rangle \right)\rangle$ denotes an average performed over all the time steps of the trajectory.The principal components (PCs) are obtained by a diagonalisation of the covariance matrix C (see Equation (2)).
(2)$$C=V\Lambda V^{T}$$This results in a diagonal matrix Λ containing eigenvalues as diagonal entries and a matrix V containing the corresponding eigenvectors. If the eigenvectors are sorted such that their eigenvalues are in decreasing order, the eigenvector with the largest eigenvalues (i.e., the first PCs) accounts for the highest proportion of variance within the data. The second component is orthogonal to the first one and accounts for the second-highest proportion of variance, and so on.

(5)The extent to which the fluctuations of a system are correlated depends on the magnitude of the cross-correlation coefficient (CC*ij*). The CC*ij* of the atomic fluctuations obtained from the MD simulations (CC^PCA^) were computed using Equation (3):
(3)$${\mathrm{CC}}_{IJ}^{PCA}=\frac{\langle \Delta r_{i}^{T}\Delta r_{j}\rangle }{{\langle \Delta r_{i}^{T}\Delta r_{i}\rangle }^{1/2}{\langle \Delta r_{j}^{T}\Delta r_{j}\rangle }^{1/2}}$$
where *i* and *j* are two atoms of Cα; Δ*r*_i_ and Δ*r*_j_ are displacement vectors of *i* and *j*; $\Delta r^{T}$ denotes the transpose of a column vector. If *CC*(*ij*) = 1, the fluctuations of *i* and *j* are completely correlated.If *CC*(*ij*) = −1, the fluctuations of *i* and *j* are completely anticorrelated. If *CC*(*ij*) = 0, the fluctuations of *i* and *j* are not correlated. All snapshots were fitted using the transmembrane domain Cα as a reference before performing cross-correlation analysis.The Normal Mode Wizard (NMWiz) plugin [73] of the Visual Molecular Dynamics (VMD) 1.9.3 program [74] was used to visualise the motions along with the principal components.

(6)Clustering analysis was performed on the productive simulation time of each MD trajectory using an ensemble-based approach [28]. The first 70 ns were omitted from the analysis of Trx-fold proteins. The analysis was performed every 100 ps.The algorithm extracts representative MD conformations from a trajectory by clustering the recorded snapshots according to their Cα-atom RMSDs. The procedure for each trajectory can be described as follows: (i) a reference structure is randomly chosen in the MD conformational ensemble, and all conformations within an arbitrary cutoff *r* are removed from the ensemble; this step is repeated until no conformation remains in the ensemble, providing a set of reference structures at a distance of at least *r*; (ii) the MD conformations are grouped into *n* reference clusters based on their RMSDs from each reference structure. The cut-off was set to 2 Å for both clustered proteins or domains (Trx-fold and L-loop) to allow the comparison.

(7)Drift analysis of helices was performed on the L-loop from h-VKORC1 using the centroids (C_i_) defined for the main-chain atoms for amino acids (aas) at the top and bottom of each helix. Positions of these centroids were monitored over the MD simulations, and their coordinates were projected on the *x–z* and *y–z* planes. The geometry of the CX_1_X_2_C motif from the Trx-fold proteins was described by the distance S⋯S between two sulphur atoms from cysteine residues C37 and C40 and the dihedral angle determined as an absolute value of pseudotorsion angle S− Cα37− C40α−S.

(8)H-bonds between heavy atoms (*N*, *O*, and *S*) as potential donors/acceptors were calculated with the following geometric criteria: donor/acceptor distance cut-off was set to 3.6 Å, and the bond angle cut-off was set to 120°. Hydrophobic contacts were considered for all hydrophobic residues with side chains within a distance of 4 Å of each other.Visual inspection of the conformations and figure preparation was made with PyMOL (https://pymol.org/2/). The VMD 1.9.3 program [74] was used to prepare the protein MD animations. To visualise the motions along the principal components, the Normal Mode Wizard (NMWiz) plugin [73], which is distributed with the VMD program, was utilised.

#### 4.3.2. Advanced Methods of Analysis

(1)Metric multidimensional scaling (MDS) is an algorithm for dimension reduction and visualization; it computes an embedding of a set of points (a shape trajectory in our case) in a lower dimension space with respect to the pairwise distances (Kendall’s ones in our case) in the original set [19].The algorithm consists of a minimisation of the cost (see Equation (4)):
(4)$$\overset{}{\underset{i\neq j}{\sum }}{\left(d_{ij}-\|x_{i}-x_{j}\|\right)}^{2}$$
where $D=\left(d_{ij}\right)$ is the pairwise distance matrix, and ${\left\{x_{i}\right\}}_{i}$ are the embedded points. It can be implemented using the manifold. MDS class in Python’s scikit learn library.

(2)The Fréchet mean of a set is a point minimising the sum of squared distances to each point of the set. As an example, the Fréchet mean $\bar{T}$
of one set ${\left\{T_{i}\right\}}_{i}$ of tetrahedrons is defined as in Equation (5):(5)$$\bar{T}\in \underset{T}{argmin}\underset{i}{\sum }d{\left(T,T_{i}\right)}^{2}$$When the distance is the Euclidean distance, the Fréchet mean is no other than the classical mean we know.

(3)Kendall’s shape space of 3D triangles is isometric to the northern hemisphere of a 3D sphere of radius 12
where the equilateral triangle is at the north pole [20]. We use a planar representation of the half-sphere as a disk with the equilateral triangle at the centre by the transformation $\left(\varphi ,\theta \right)\to \left(r=\sin\left(\theta \right),\varphi \right)$ from the spherical coordinates to the polar coordinates. Each 3D triangle, up to translation rotation and scaling, is represented by a unique point of the disk.

## Figures and Tables

**Figure 1 ijms-22-00802-f001:**
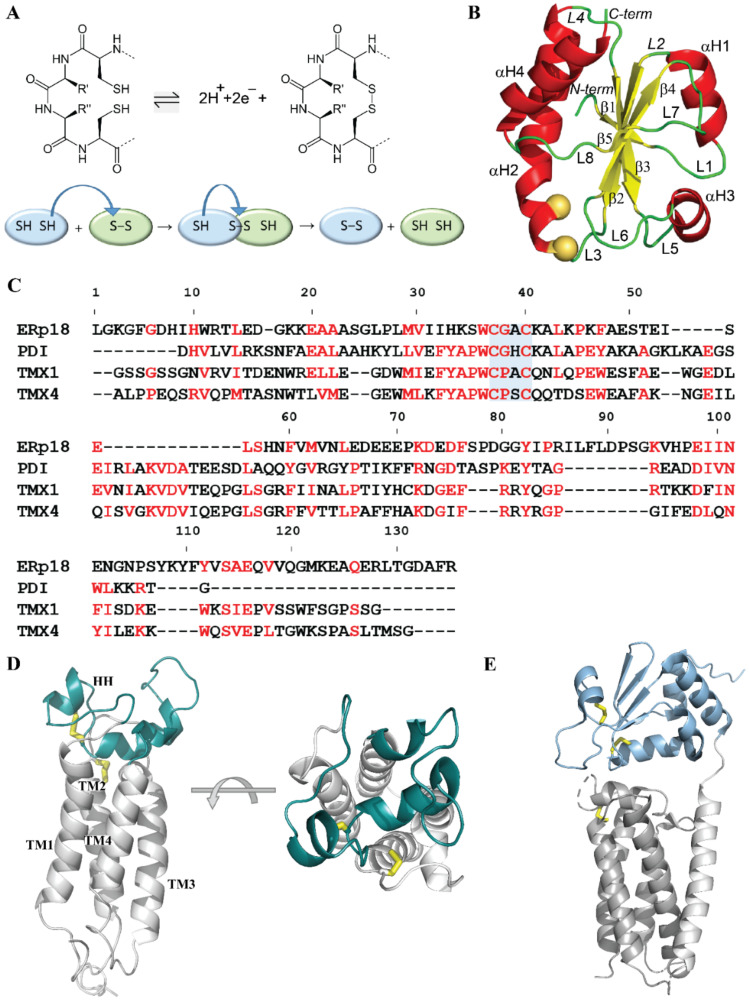
Thioredoxin-fold protein as a physiological reductant of human vitamin K epoxide reductase complex 1 (hVKORC1). (**A**) Oxidation of the two cysteine residues in the CX_1_X_2_C motif of Trx-fold proteins forms a disulphide bond, a process associated with the loss of two hydrogen atoms and, hence, two electrons (top). Mechanism of disulphide exchange between Trx and a target (bottom). The *H*-donor enzyme and a target are coloured in blue and green, respectively. (**B**) The Trx-fold is illustrated using the X-ray structure of human PDI deposed in PDB [5] (PDB ID: 4ekz). The protein is shown as red ribbons, with two cysteine residues from the CX_1_X_2_C motif as yellow balls. The four α-helices (in red), five β-strands (in yellow) and eight loops (in green) are numbered. (**C**) Comparison of the sequences of Trx-fold proteins ERp18, PDI, Tmx1 and Tmx4. Sequences were aligned on ERp18, having the most elongated sequence with ESPript3 (http://espript.ibcp.fr/). The solution with the best score is shown. The residues are coloured according to the consensus values: red indicates strict identity or similarity, while nonconserved residues are in black. Blue highlights the CX_1_X_2_C motif. (**D**) Ribbon diagram of the 3D human VKORC1 model in its inactive state showed in two orthogonal projections. The L-loop is shown in the colour teal, while disulphide bridges formed by cysteine residues C43—C51 and C132—C135 are drawn as yellow sticks. The transmembrane helices (TM) are numbered as in [6]. (**E**) The structure of VKOR from Synechococcus sp (bVKOR; ID PDB: 4nv5) is visualised using ribbons. The structural fragments that have sequences most similar to hVKORC1 and the Trx-like domain are shown in dark grey and light blue, respectively. The disulphide bridges formed by cysteine residues in Trx-like and VKOR-like domains are drawn as yellow sticks.

**Figure 2 ijms-22-00802-f002:**
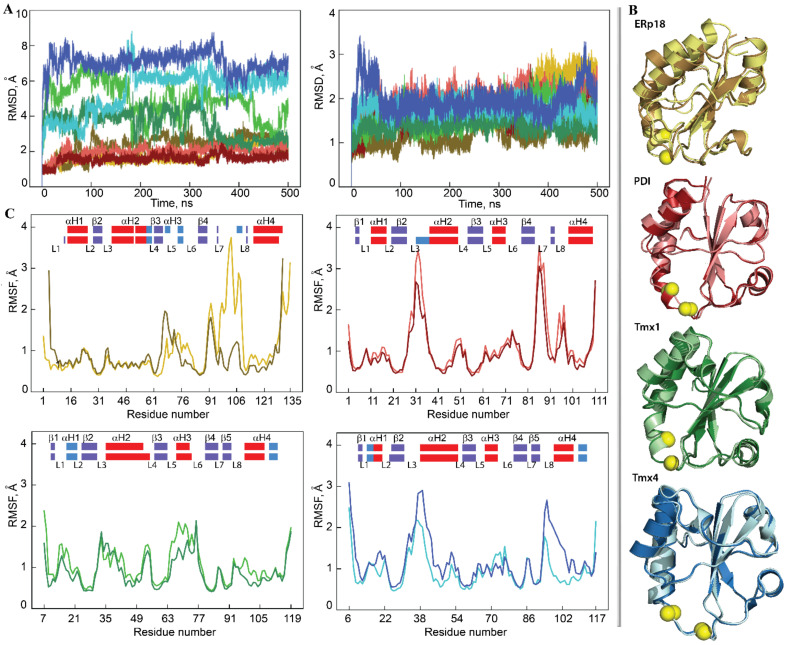
Characterisation of the MD simulations for the four Trx-fold proteins ERp18, PDI, Tmx1 and Tmx4. (**A**) RMSDs from the initial coordinates computed for all Cα-atoms (right) in each protein after fitting to initial conformation. (**B**) The superimposed average structures of each protein over replicas **1** and **2**. Cysteine residues are shown as yellow balls. RMSD values of 0.5, 0.4, 0.3 and 0.4 Å in Erp18, PDI, Tmx1 and Tmx4, respectively. (**C**) RMSFs computed for the Cα-atoms using RMSF amplitude values less than 4 Å for the MD conformation of each protein after fitting to the initial conformation. Highly fluctuating residues (3, 6 and 5 in ERP18, Tmx1 and Tmx4, respectively) were excluded from the RMSD computation. In the insert, the secondary structures—αH- (red), 3_10_-helices (light blue) and β-strands (dark blue)—were assigned for a mean conformation of every MD trajectory, **1** (top) and **2** (bottom), of each protein and were labelled as in the crystallographic structure of human PDI. (**A**–**C**) Proteins are distinguished by colour (first/second replicas): ERp18 (yellow/brown), PDI (light/dark red), Tmx1 (light/dark green) and Tmx4 (light/dark blue). The numbering of the residues in each Trx-fold protein is arbitrary and starts from the first amino acid in the 3D model.

**Figure 3 ijms-22-00802-f003:**
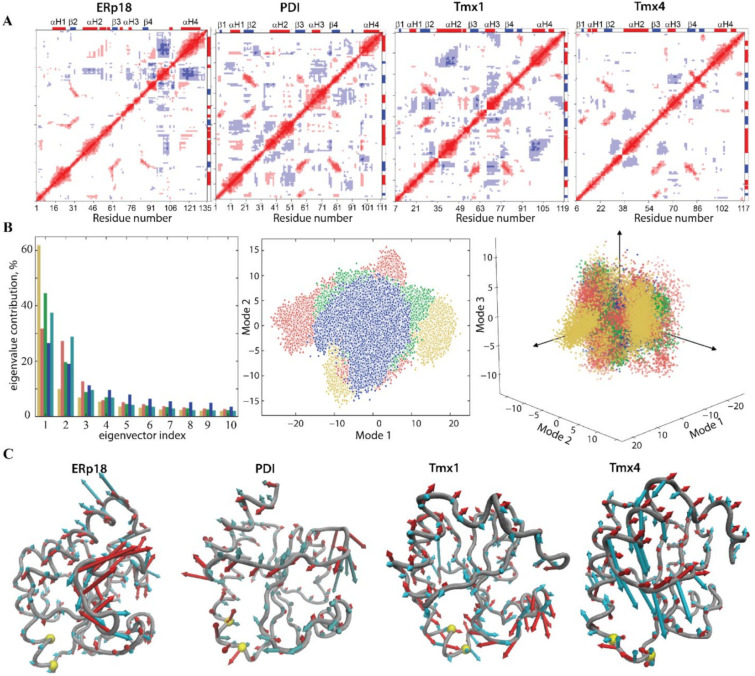
Intrinsic motion in the Trx-folded proteins and its interdependence. (**A**) Inter-residue cross-correlation maps computed for the Cα-atom pairs of ERp18, PDI, Tmx1 and Tmx4 after the fitting procedure. Secondary structure projected onto the protein sequences (α-helix/β-strand in red/blue) is shown at the border of matrices. Correlated (positive) and anticorrelated (negative) motions between Cα-atom pairs are shown as a red–blue gradient. (**B**) The PCA modes calculated for each protein after least-square fitting of the MD conformations to the average conformation as a reference. The bar chart gives the eigenvalue spectra in descending order for the first 10 modes (left). Projection of ERp18, PDI, Tmx1 and Tmx4 MD conformations with the principal component (PC) in 2D (middle) and 3D subspaces (right). MD conformations were taken every 100 ps (2D) and 10 ps (3D). The protein data is referenced by colour—ERp18 (dark yellow), PDI (brown), Tmx1 (green) and Tmx4 (dark blue and light blue for two replicas). (**C**) Collective motions characterised by the first two PCA modes. Atomic components in PCA modes 1–2 are drawn as red (1st mode) and cyan (2nd mode) arrows projected on a tube representation of each protein. For clarity, only motion with an amplitude ≥2 Å is represented. Cysteine residues are shown as yellow balls. All computations were performed on the Cα-atoms with RMSF fluctuations less than 4 Å for each protein after fitting on the initial conformation.

**Figure 4 ijms-22-00802-f004:**
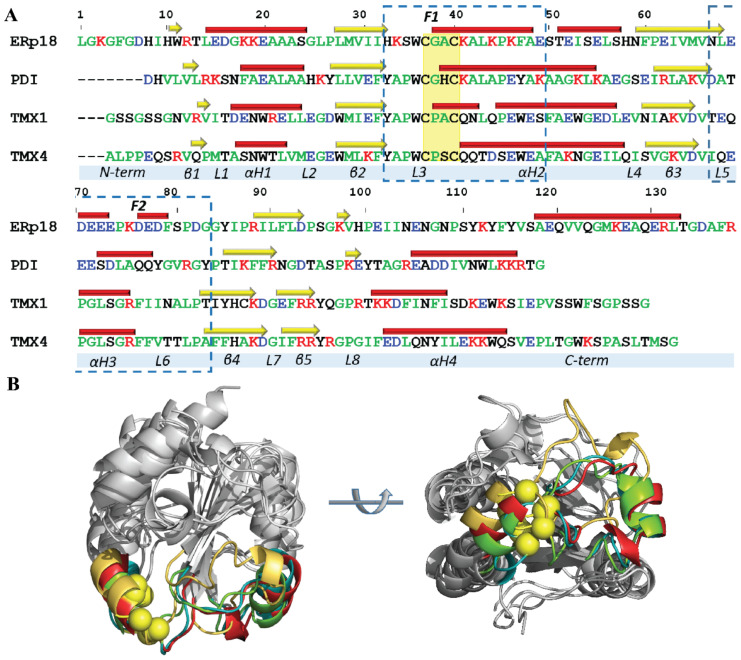
Sequence and folding of Trx-like proteins. (**A**) Alignment of the sequences and the secondary structure assigned to a mean conformation of the concatenated trajectory of each studied protein. Residues are coloured according to their properties—the positively and negatively charged residues are in red and blue, respectively; the hydrophobic residues are in green; the polar and amphipathic residues are in black; the CX_1_X_2_C motif is highlighted by a yellow background. The α-helices and β-strands are shown as red batons and yellow arrows, respectively. Secondary structure labelling is shown below the Tmx4 sequence. (**B**) The superimposed 3D structures of the Trx-fold proteins are shown in two orthogonal projections. The proteins are drawn as ribbons, with the cysteine residue as yellow balls. The **F1** and **F2** regions (and secondary structure labels) that are potentially involved in target recognition and/or the electron transfer reaction are outlined by dashed lines in (**A**) and differentiated by colour in (**B**) to distinguish between the proteins: ERp18 (dark yellow), PDI (red), Tmx1 (green) and Tmx4 (dark blue).

**Figure 5 ijms-22-00802-f005:**
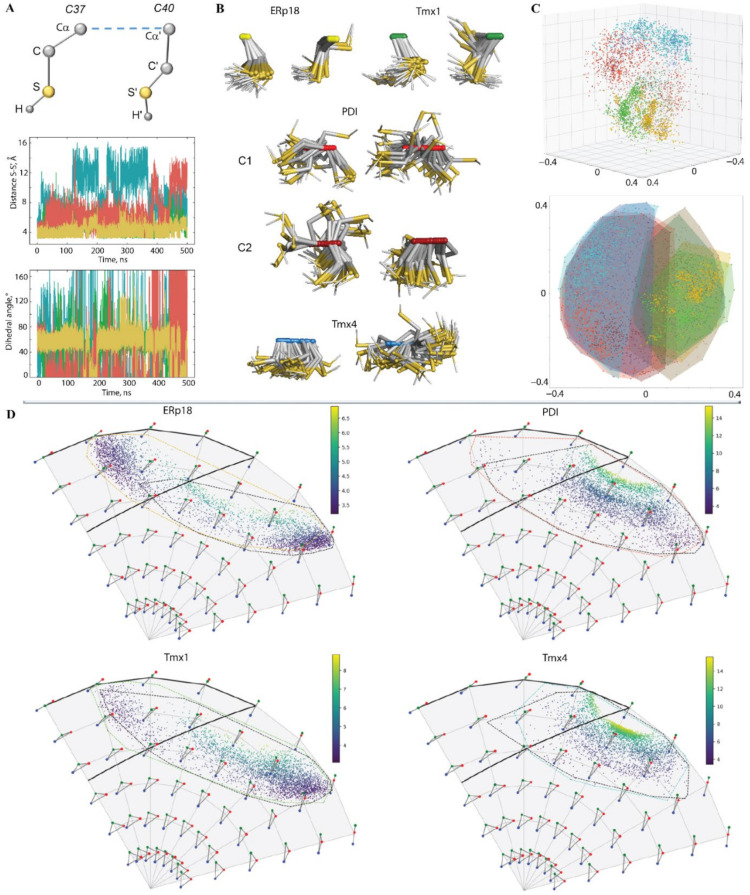
The CX_1_X_2_C motif geometries for ERp18, PDI, Tmx1 and Tmx4. (**A**) Geometry of the CX_1_X_2_C motif (left) is described by distance S⋯S’ (middle) and dihedral angle (right), determined as an absolute value of the pseudo torsion angle S−Cα(C37)−Cα’(C40)−S’. Only one replica **2** is shown. (**B**) Superposition of the thiol groups (Cα-C-S-H) from the CX_1_X_2_C motif of each protein is shown for either only one MD trajectory (ERP18, Tmx1 and Tmx4) or for both (PDI). Samples were taken for each 100-ns frame. (**C**) Multidimensional scaling (MDS) in 2D and 3D on the set of S-C-C-S tetrahedrons. Embedded points have been coloured according to the partner and replica they belong to. (**D**) Evolution of the shape of the triangles S-H⋯S on Kendall’s disk of 3D triangles; each data point is coloured according to the S⋯S distance. Representative triangles are regularly sampled on the disk. The thick black line delimits the area of conformations favouring H-bond interaction. The dashed areas are contouring subpopulations according to the S-atom being the H-donor.

**Figure 6 ijms-22-00802-f006:**
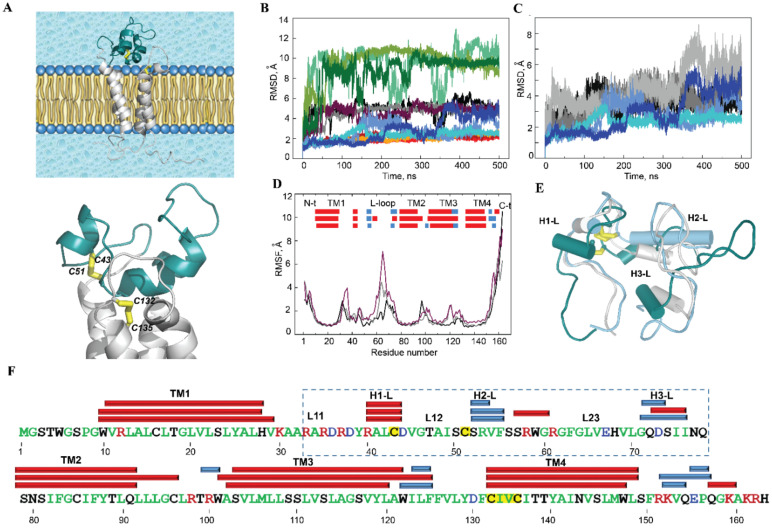
hVKORC1 in its inactive state and its conventional MD simulations. (**A**) 3D model of hVKORC1 in its inactive state; it was inserted into the membrane (top) and zoomed in on the L-loop (bottom). The L-loop is highlighted by the colour teal; disulphide bridges formed by cysteine residues C43-C51 and C132-C135 are drawn as yellow sticks. Transmembrane helices (TMs) are numbered as in [6]. (**B**,**C**) RMSDs computed for each MD trajectory (replicas **1–3**) from initial coordinates (at *t* = 0 ns, the same for all replicas) on the Cα-atoms of full-length hVKORC1 (in black, grey and rose brown), of the transmembrane domain (in orange, red and grenadine), of the L-loop (in clear aqua, bleu and navy) and of the *N*- and *C*-terminals (in teal, green and deep green) after fitting to the initial conformation of the respective fragment (**B**); of the L-loop (i) after fitting to its initial conformation (clear aqua, blue and navy blue) and (ii) after fitting of the protein coordinates to the initial conformation of the TMD (black, grey and silver) (**C**). (**D**) RMSFs computed for Cα-atoms of the MD conformations (replicas **1**–**3**) after fitting to the initial conformation (at *t* = 0 ns, the same for all replicas; in black, grey and rose brown). In the insert, the folded secondary structures, αH- (red) and 3_10_-helices (blue), were assigned for a mean conformation of each MD trajectory. (**E**) Superimposition of the L-loop conformations picked from replica **3** at 150 (grey), 250 (light blue) and 375 ns (deep teal). (**F**) The hVKORC1 sequence (Q9BQB6) and the secondary structure assignment for a mean conformation over each MD trajectory. Residues are coloured according to their properties: positively and negatively charged residues are in red and blue, respectively; hydrophobic residues are in green; polar and amphipathic residues are in black; residues C43, C51 and the CX_1_X_2_C motif are highlighted by a yellow background. α- and 3_10_-helices are shown as red and blue batons, respectively. Secondary structure labelling is shown above the VKORC1 sequence. The L-loop sequence is surrounded by dashed lines.

**Figure 7 ijms-22-00802-f007:**
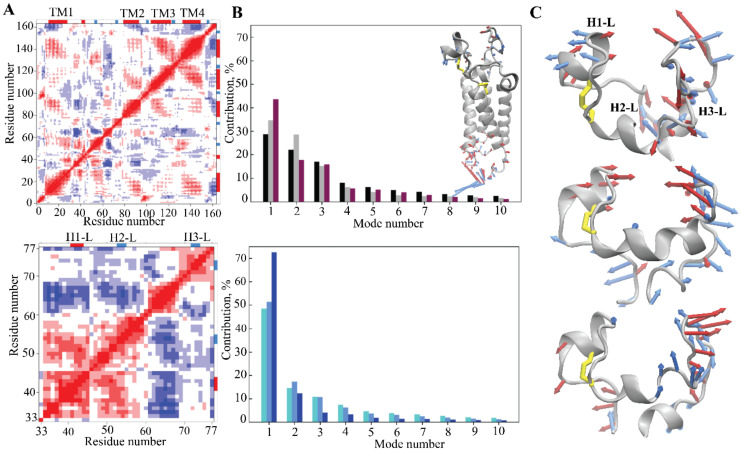
Intrinsic motion of hVKORC1 and its L-loop. (**A**) The inter-residue cross-correlation map computed for the Cα-atom pairs after fitting to the respective first conformation (*t* = 0 ns) of the full-length hVKORC1 (top) and the L-loop (bottom) is shown for the three replicas. Correlated (positive) and anticorrelated (negative) motions between the Cα-atom pairs are shown as a red–blue gradient. (**B**) The PCA modes of the full-length hVKORC1 (top) and the L-loop (bottom), calculated for each MD trajectory after least-square fitting of the MD conformations to the average conformation of the respective domain as a reference. The bar plot gives the eigenvalue spectra in descending order for the first 10 modes. The data for replicas **1**–**3** are coloured black, grey and rose brown, respectively, while, for the full-length hVKORC1, the colouring is clear aqua, blue and navy blue for the L-loop. (**C**) Atomic components in the first PCA modes of the L-loop are drawn as red (1st mode) and blue (2nd mode) arrows projected onto the respective average structure from replicas **1** (top), **2** (middle) and **3** (bottom). Only motion with an amplitude ≥2Å is shown. The S-S bridge of hVKORC1 is shown using yellow sticks.

**Figure 8 ijms-22-00802-f008:**
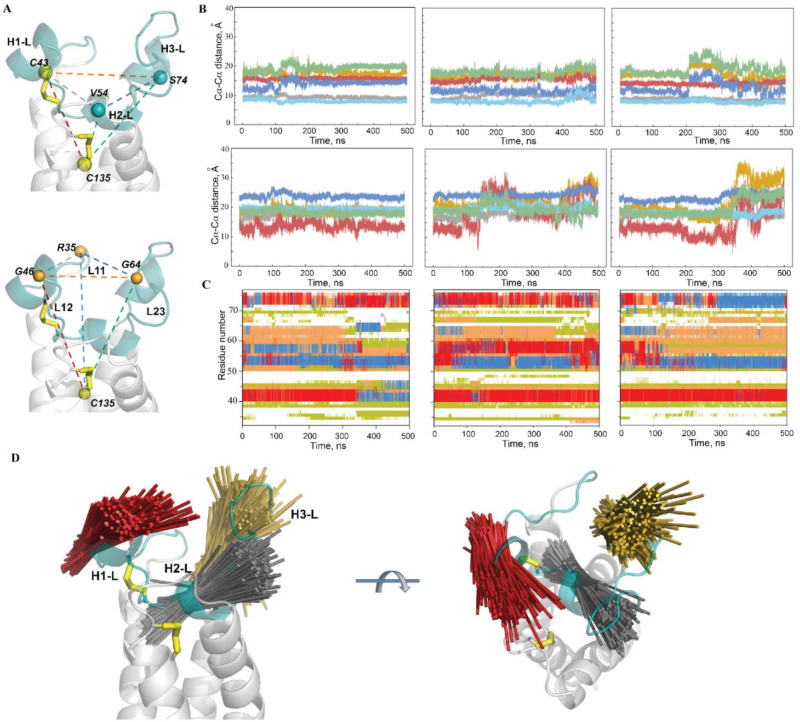
Geometry and folding of the L-loop from hVKORC1 in its inactive state. (**A**) Two tetrahedrons, **T1**—defined for the Cα-atom of C135 and for the midpoint residues of each L-loop helix, and **T2**—defined for the Cα-atom of C135 and for the most fluctuating residues (with the greatest RMSF values), from the L-loop linkers. (**B**) Distances between each pair of Cα-atoms from the tetrahedrons **T1** (top) and **T2** (bottom) over each MD trajectory. The distance curves and the edges of a tetrahedron are coloured similarly. (**C**) The time-dependent evolution of the secondary structure of each residue, as assigned by the Define Secondary Structure of Proteins (DSSP) method: α-helix is in red, 3_10_-helix is in blue, turn is in orange and bend is in dark yellow. (**D**) Drift of the L-loop helices observed over the MD simulations (concatenated trajectory, sampled every 100 ps). Superimposed axes of helices from the L-loop are covered on the randomly chosen conformation of hVKORC1 in two orthogonal projections. The axis of each helix is defined as a line connecting the two centroids assigned for the first and the last residues.

**Figure 9 ijms-22-00802-f009:**
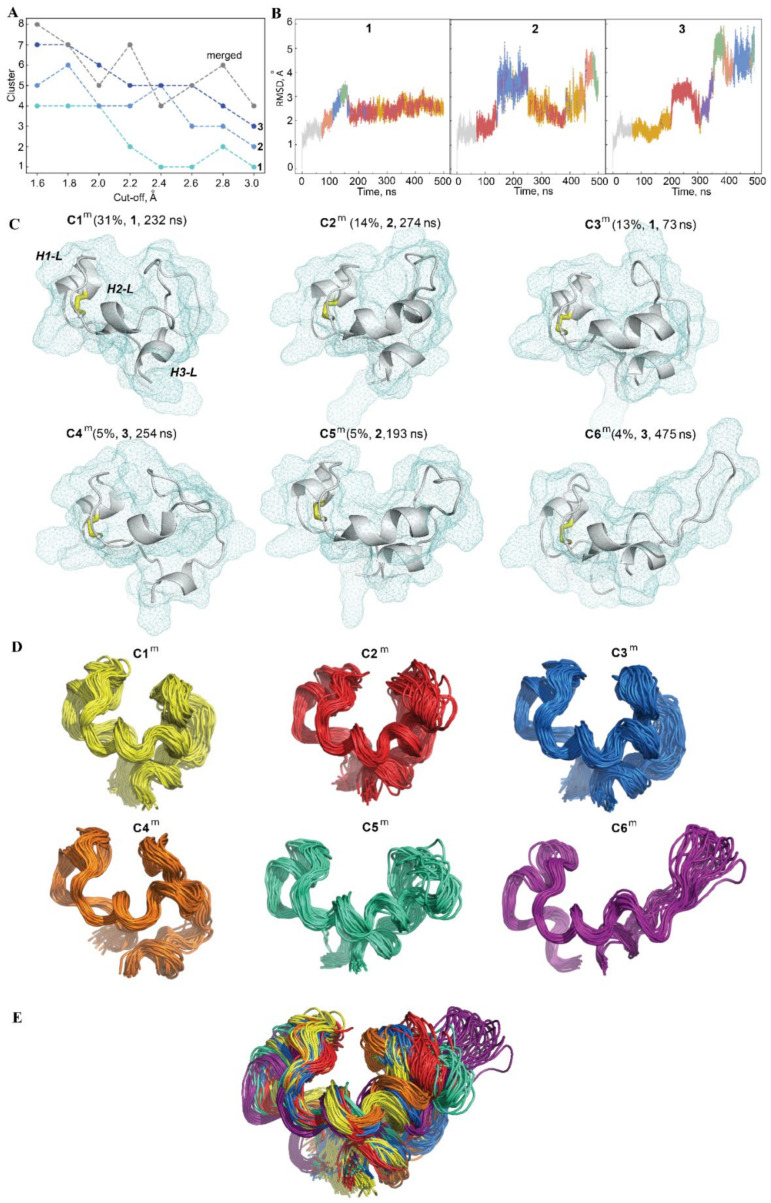
Ensemble-based clustering of L-loop MD conformations. (**A**) Number of clusters obtained for each MD trajectory (**1**, **2** and **3**) and the concatenated trajectory. The first 70 ns of every trajectory was omitted from the computation. Clustering was performed on each 10-ps frame of every trajectory using cut-off values that varied from 1.6 to 3.0 Å, with a step of 0.2 Å. (**B**) Location of the MD conformations grouped in clusters, with a cut-off of 2.0 Å for the RMSD curves of trajectories **1**–**3**. Clusters C1–C6 are arbitrarily distinguished by colours in each trajectory: orange (C1), red (C2), blue (C3), rose (C4), green (C5) and violet (C6). (**C**) Representative conformations of the L-loop from clusters (C^m^) with population ≥4%, obtained with a cut-off of 2.0 Å for the merged trajectory. The L-loop is shown as ribbons with a meshed surface, with disulphide bridges C43–C51 drawn as yellow sticks. The L-loop surface is displayed as meshed contours. The population of each cluster is given in brackets (in %), together with the replica number (in the bold) and the time (in ns) over which the representative conformation was recorded within a replica. (**D**) Conformations of the L-loop (taken every 100 frames) of each cluster (C^m^) of the merged trajectory, and (**E**) superposed conformations from the C1^m^–C6^m^ clusters. In (**D**,**E**), the L-loop is drawn as a tube.

**Figure 10 ijms-22-00802-f010:**
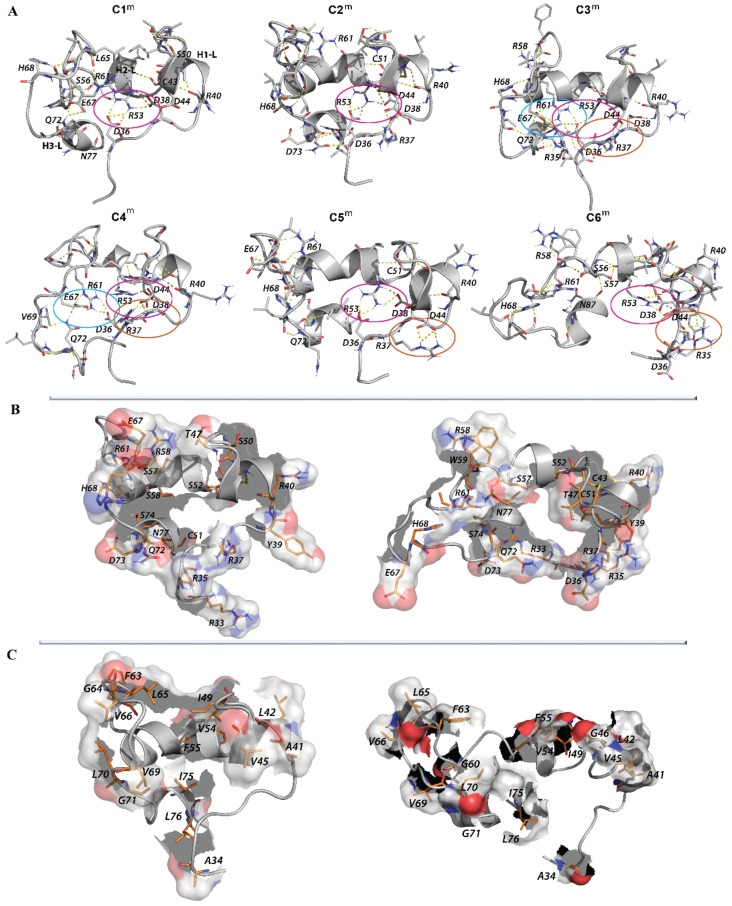
Interacting residues in L-loop conformations. (**A**) Intraloop H-bond interactions in the L-loop conformations from clusters C1^m^–C6^m^. H-bonds D-H⋯A (D⋯A < 3.6 Å, ∠DHA ≥ 120°), where D and A are *H*-donor and *H*-acceptor (O/N) atoms, were analysed in a representative conformation from each cluster of the merged trajectories. Interactions that stabilised the helices were not considered. The L-loop is shown as ribbons, with the interacting residues as sticks and H-bond traces as dashed lines. Common *H*-bonding motifs are encircled by magenta (at R53), blue (at R61) and orange (at R37). The most characteristic donor and acceptor groups are labelled. *N*, *O* and *C* atoms are in blue, red and grey, respectively. (**B**) Charged and polar residues protruding from the L-loop. (**C**) Hydrophobic residues protruding from of the L-loop. The L-loop is shown as ribbons, with the residues exposed to the solvent displayed as sticks with a space-filling encounter. In (**B**,**C**), the *N*, *O* and *C* atoms are in blue, red and orange, respectively.

**Figure 11 ijms-22-00802-f011:**
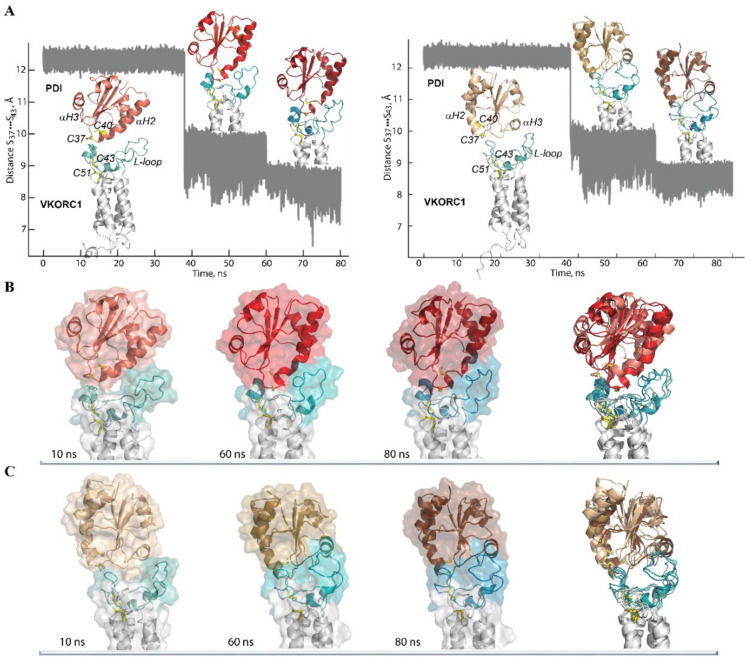
Modelling of human PDI–VKORC1 complex. (**A**) MD simulations of 3D model PDI–VKORC1 complexes were performed, with gradually diminished distance (from 12.5 to 8.0 Å) between the sulphur (S) atoms of C37 from PDI and of C43 from the L-loop of hVKORC1. PDI has two orientations with respect to VKORC1, with **F1** (**Model 1**, left) and **F2** (**Model 2**, right) positioned above the middle of the L-loop surface. Both models of the PDI–VKORC1 complex are shown as snapshots taken at *t* = 10, 60 and 80 ns, with different S⋯S distances. The reference residues and fragments are labelled. (**B**,**C**) Conformations of the PDI–VKORC1 complex, with two different PDI orientations, chosen at *t* = 10, 60 and 80 ns, and their superposition at all three times. In (**A**–**C**), the proteins are depicted as ribbons or as ribbons and surfaces and are distinguished by colour: a red palette was used for PDI and a cyan palette for hVKORC1, both nuanced by the tonality from light to dark to distinguish the conformations chosen at *t* = 10, 60 and 80 ns.

**Figure 12 ijms-22-00802-f012:**
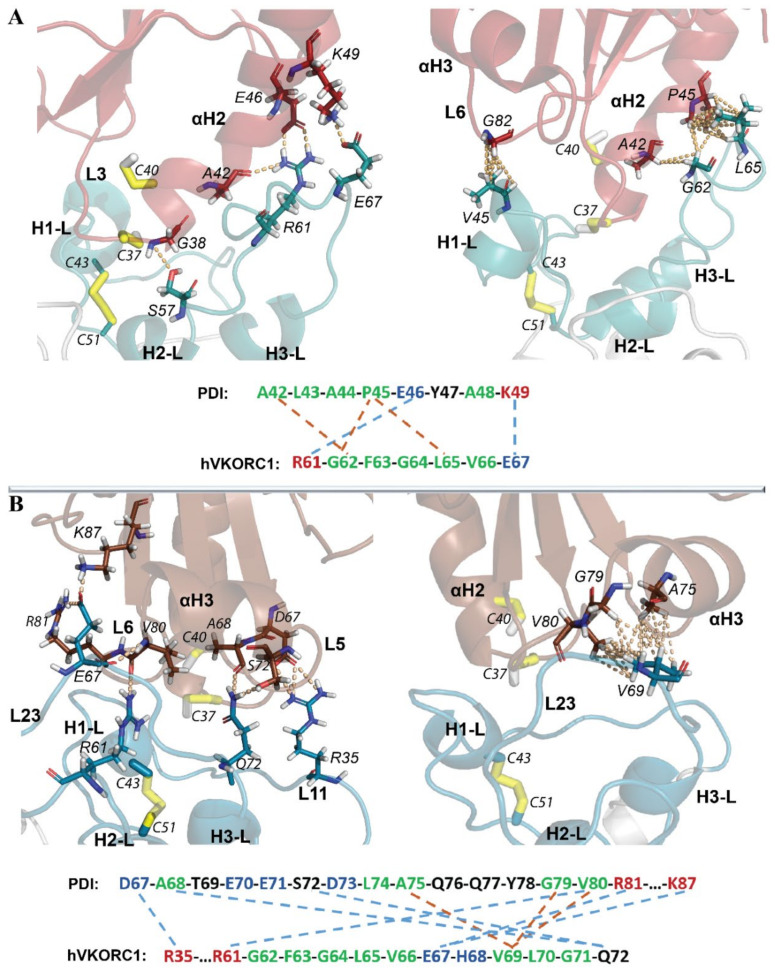
Intermolecular contacts at the interface between PDI and hVKORC1 in two models of the PDI–hVKORC1 complex. The intermolecular H-bonds and hydrophobic contacts between PDI and VKORC1 in Model 1 (**A**, top) and Model 2 (**B**, top). (**A**,**B**) The proteins are shown as coloured ribbons: PDI in red and brown and VKORC1 in cyan (L-loop), with the interacting residues and thiol groups as sticks. The contacts are indicated by dashed lines: H-bonds in yellow and hydrophobic contacts in salmon. The structural fragments and residues participating in the contacts are labelled. Analysis of the intermolecular contacts was performed on conformations taken at *t* = 80 ns. (**A**,**B**) A pattern of H-bond (in blue) and hydrophobic (in orange) contacts between the PDI and hVKORC1 residues (bottom). Residues are coloured according to their properties: the positively and negatively charged residues are in red and blue, respectively; the hydrophobic residues are in green; the polar and amphipathic residues are in black.

## Data Availability

The numerical model simulations upon which this study is based are too large to archive or to transfer. Instead, we provide all the information needed to replicate the simulations. The models coordinates are available from L. Tchertanov at ENS Paris-Saclay.

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
