# Peer review of "Identification of the Primary Factors Determining the Specificity of Human VKORC1 Recognition by Thioredoxin-Fold Proteins"

_ijms, 2021, doi:10.3390/ijms22020802_

Round 1

Reviewer 1 Report

In this manuscript by Stolyarchuk et al., the authors use MD simulations to study the interaction between human VKORC1 and the redoxin PDI. They first examine the conformations of VKORC1 and 4 redoxins in isolation to identify important structural features. They then build a model for the complex between PDI and VKORC1 that allow them to identify residues that mediate the interaction.

Although the data seem reasonable and the authors did a good job in examining multiple aspects of the simulations and drew interesting conclusions from them, the manuscript requires extended editing to increase clarity and reach a larger public.

Major comments:

-The text is very long and some parts, while interesting to discuss, could go in supplementary material (for example, intrinsic movement of redoxins and intra-L-loop interactions could be shortened to focus on the interactions interfaces). 

-the justification of the importance of this work is missing in the introduction and only partially appears in the beginning of the discussion. I suggest reorganizing.

-I didn't understand how PDI was selected as the probable biological redoxin for VKORC1, though it seemed to be an important aspect of this study... It is partially discussed in the discussion part but should be introduced in the results as rationale.

- similarly, the rationale for the 2 models is not clearly explained.

Overall, the details of the work were correctly described but the big picture was confused/missing.

Minor comments:

- I recommend using the direct, rather than undirect form when the results emerge directly from this study (e.g l 881: "To attack the problem, it is first suggested that..." is unclear, is it "it has been suggested (by others) that..." or " we suggest/show that ...")  

- Some parts of the manuscript are very difficult to understand . As an example, l936-942. This part is very opaque, especially the justification that "F1 of PDI and the targeted S56-R61 segment of VKORC1 have a similar structural propriety" given the fact that they are both disordered... Similarly "the best structural compatibility from the point of view of their concerted structural reorganization" is unclear.

- wrong/overuse of definite and indefinite articles (e.g. l97-98; "namely the protein disulfide isomerase (PDI), the endoplasmic reticulum oxidoreductase (ERp18)...")

Author Response

Responses to Reviewer #1:

The Authors thank Reviewer 1 for critical reading of our manuscript. Our responses are supplied after each concern.

In this manuscript by Stolyarchuk et al., the authors use MD simulations to study the interaction between human VKORC1 and the redoxin PDI. They first examine the conformations of VKORC1 and 4 redoxins in isolation to identify important structural features. They then build a model for the complex between PDI and VKORC1 that allow them to identify residues that mediate the interaction.

Although the data seem reasonable and the authors did a good job in examining multiple aspects of the simulations and drew interesting conclusions from them, the manuscript requires extended editing to increase clarity and reach a larger public.

Response: Thank you very much for the positive comments of the manuscript and critical remarks.

Major comments:

-The text is very long and some parts, while interesting to discuss, could go in supplementary material (for example, intrinsic movement of redoxins and intra-L-loop interactions could be shortened to focus on the interactions interfaces).

Response: We do not agree with the Referee's suggestion to reduce the text, move a few figures to the additional material, and focus on only the interactions interface. We cannot know in advance which properties of protein (structure, dynamics or just the sequence composition at the interface) will be the determining factors for the recognition of its partner. Moreover, the studied proteins (Trxs and VKORC1) are allosterically regulated systems with a very variable surface area highly depending on the protein folding and intrinsic dynamics which in turn are stabilized by interactions, and therefore, a complete presentation of obtained results is absolutely essential.

-the justification of the importance of this work is missing in the introduction and only partially appears in the beginning of the discussion. I suggest reorganizing.

Response: The required reorganizing was made (lines 85-90 and 870-876).

-I didn't understand how PDI was selected as the probable biological redoxin for VKORC1, though it seemed to be an important aspect of this study... It is partially discussed in the discussion part but should be introduced in the results as rationale.

Response: The choice of PDI as the probable biological partner for hVKORC1 was made only after the detailed characterization of the hVKORC1 properties. First we identified the most probable ‘interacting’ fragment on VKORC1 (lines 906-908: “the S56-R61 segment, a part of the more extended S53-N77 segment, is a platform for recognition of a protein partner”). Since this segment is an intrinsically disordered region (IDR), its ‘best’ partner should be a protein having the similar properties (see the cited literature). Only in PDI the CX1X2C motif, the principal actor in electron–transfer reaction, is also an IDR. The transient state of CX1X2C motif is explained at lines 199-201; 345-347; 366-368.

The role of the disordered fragments in formation of molecular complexes is now well-recognized (see the cited references, and also Wallmann, A.; Kesten, C. Common Functions of Disordered Proteins across Evolutionary Distant Organisms. Int. J. Mol. Sci. 2020, 21, 2105). Additional example, NMR studies and isothermal calorimetry experiments of intrinsically disordered complexes have provided direct evidence for the role of structural, dynamic, and frustrational plasticity in interactions between intrinsically disordered proteins (P Jemth, Elin Karlsson, B Vögeli, B Guzovsky, E Andersson, G Hultqvist, J Dogan, P Güntert, R Riek, C Chi Structure and dynamics conspire in the evolution of affinity between intrinsically disordered proteins. Science Advances 24 Oct 2018: eaau4130). Thus, three historical complexes demonstrate how an extensive reorganization of structure and interactions, rather than specific side-chain substitutions in an intact structure, can lead to a higher affinity in a protein-protein complex. Another example, recognition of signaling proteins by receptors tyrosine kinase in which the kinase activation loop, the juxtamembrane region, and the kinase insert domain (all are intrinsically disordered regions) recruit signalling effectors through pTyr binding of the SH2 and PTB domains (also partially disordered) of signalling effectors (Wagner MJ, Stacey MM, Liu BA, Pawson T. Molecular mechanisms of SH2- and PTB-domain-containing proteins in receptor tyrosine kinase signaling. Cold Spring Harb Perspect Biol. 2013;5(12):a008987. doi: 10.1101/cshperspect.a008987; Ren, S., Uversky, V.N., Chen, Z. et al. Short Linear Motifs recognized by SH2, SH3 and Ser/Thr Kinase domains are conserved in disordered protein regions. BMC Genomics 9, S26 (2008). https://doi.org/10.1186/1471-2164-9-S2-S26).

- similarly, the rationale for the 2 models is not clearly explained.

Response: See lines 731- 768 and 966-976.

Overall, the details of the work were correctly described but the big picture was confused/missing.

Response: We are agreeing with the Reviewer that paper is difficult to a reading but it is not ‘confused/missing’. Our work comprises 3 logically-related but topically independent parts: (1) comparison between four Trx proteins, (2) characterization of the inactive state of hVKORC1, and (3) modelling of PDI-hVKORC1 complex. Each of these parts may be published as a separate paper of a ‘normal’ size (12-15 pages and 4 figures). Since we have set ourselves a more complex problem than describing each type of protein, we have combined our results to travel the path from a purely descriptive content of each protein to the molecular complex through prediction and modelling. Probably, the text of manuscript may be reorganized to be more fluent by a mixing of the results and discussion within each part, but we respected the format required by the Journal (IJMS) for the manuscript.

Minor comments:

- I recommend using the direct, rather than undirect form when the results emerge directly from this study (e.g l 881: "To attack the problem, it is first suggested that..." is unclear, is it "it has been suggested (by others) that..." or " we suggest/show that ...")  

Response: The required change was made (lines 881 and 909) and elsewhere when needed.

- Some parts of the manuscript are very difficult to understand . As an example, l936-942. This part is very opaque, especially the justification that "F1 of PDI and the targeted S56-R61 segment of VKORC1 have a similar structural propriety" given the fact that they are both disordered... Similarly "the best structural compatibility from the point of view of their concerted structural reorganization" is unclear.

Response:  The text was modified (lines 937-940) to clarify the sentence.

- wrong/overuse of definite and indefinite articles (e.g. l97-98; "namely the protein disulfide isomerase (PDI), the endoplasmic reticulum oxidoreductase (ERp18)...")

Response: The required corrections were made and elsewhere when needed. In particular, on the lines: 15, 19-21, 24, 40, 55, 71, 78, 83, 93-95, 102-104, 106, 108, 111-113, 116, 125-126, 130, 138, 150-152, 155, 162, 177, 179, 182-183, 196, 200, 206-207, 229, 233, 235, 246, 259, 266, 279, 307, 311, 320, 340, 389-390, 393, 404, 408-409, 411, 415, 423, 426, 435, 444, 451, 457-462, 466-468, 471-473, 475-477, 479, 486, 496, 508-509, 511, 516, 519, 548, 564, 572, 592, 600-602, 638-639, 653, 658, 666, 697, 712, 720, 731, 735-736, 744, 782, 794, 817, 819, 822, 825, 832-833, 837, 840-842, 850, 854, 856, 858-859, 862, 864-866, 877-879, 881-884, 886, 889-890, 892-894, 899, 902, 904-905, 907, 909, 912, 917, 920, 929, 931-932, 935-937, 944, 953, 956, 959, 962, 966, 969, 979, 981, 983-986, 993, 997-1001, 1011, 1013-1014, 1016, 1028, 1030-1031, 1033, 1039, 1045-1050, 1052, 1057, 1060, 1064, 1077, 1079, 1090, 1111-1113, 1116, 1123, 1145, 1147, 1149, 1151, 1159, 1167, 1180, 1199, 1207, 1216, 1221, 1223-1224, 1247, 1250.

Reviewer 2 Report

The manuscript Identification of the Primary Factors Determining the Specificity of the human VKORC1 Recognition by Thioredoxin-fold Proteins by Stolyarchuk et. al is interesting.

They have provided a comparative in silico analysis concentrating similarity and divergence of redoxins in their sequence, secondary and tertiary structure, dynamics, intra-protein interactions and composition of the surface exposed to the human vitamin K epoxide reductase with four redoxins and in particular the recognition mechanisms between Trx and the target hVKORC1. The MD simulation and the intrinsic motion are presented clearly in addition to the CX1X2C motif. As authors mentioned the proposed results and prediction would be very key for confirming the results and motivating for the biologist to validate experimentally and confirming the redox chemistry underyling the vital process.

Author Response

Response to Reviewer #2:

The manuscript Identification of the Primary Factors Determining the Specificity of the human VKORC1 Recognition by Thioredoxin-fold Proteins by Stolyarchuk et. al is interesting.

They have provided a comparative in silico analysis concentrating similarity and divergence of redoxins in their sequence, secondary and tertiary structure, dynamics, intra-protein interactions and composition of the surface exposed to the human vitamin K epoxide reductase with four redoxins and in particular the recognition mechanisms between Trx and the target hVKORC1. The MD simulation and the intrinsic motion are presented clearly in addition to the CX1X2C motif. As authors mentioned the proposed results and prediction would be very key for confirming the results and motivating for the biologist to validate experimentally and confirming the redox chemistry underyling the vital process.

Response: Thank you very much for the positive comments of the manuscript.